# Peripheral apoptosis and limited clonal deletion during physiologic murine B lymphocyte development

Mikala JoAnn Simpson[1,4], Anna Minh Newen[1,4], Christopher McNees[1], Sukriti Sharma[1], Dylan Pfannenstiel[1], Thomas Moyer[2], David Stephany[2], Iyadh Douagi [2], Qiao Wang[3] & Christian Thomas Mayer [1]✉

Self-reactive and polyreactive B cells generated during B cell development are silenced by either apoptosis, clonal deletion, receptor editing or anergy to avoid autoimmunity. The specific contribution of apoptosis to normal B cell development and self-tolerance is incompletely understood. Here, we quantify self-reactivity, polyreactivity and apoptosis during physiologic B lymphocyte development. Self-reactivity and polyreactivity are most abundant in early immature B cells and diminish significantly during maturation within the bone marrow. Minimal apoptosis still occurs at this site, however B cell receptors cloned from apoptotic B cells show comparable self-reactivity to that of viable cells. Apoptosis increases dramatically only following immature B cells leaving the bone marrow sinusoids, but above 90% of cloned apoptotic transitional B cells are not self-reactive/polyreactive. Our data suggests that an apoptosis-independent mechanism, such as receptor editing, removes most self-reactive B cells in the bone marrow. Mechanistically, lack of survival signaling rather than clonal deletion appears to be the underpinning cause of apoptosis in most transitional B cells in the periphery.

Antibody diversity is generated by variable, diversity, and joining (V(D)J) recombination during B cell development in the bone marrow and during somatic hypermutation in germinal centers (GC). During B cell development, pro-B cells (Fr. B/C) first recombine the immunoglobulin heavy chain (IgH). Successful IgH rearrangement signals the proliferation of large pre-B cells (Fr. C′) before resting small pre-B cells (Fr. D) begin VJ rearrangements at the immunoglobulin light chain (IgL) loci Igκ and Igλ[1]. Some immature B cells (Fr. E) expressing the fully rearranged surface IgM B cell receptor (BCR) leave the bone marrow via sinusoids and enter the blood circulation as transitional 1 (T1) B cells. In the spleen, signaling by the BCR and BAFF receptor (BAFF-R) are essential for B cell survival and differentiation into transitional 2 (T2), mature follicular (FO) and marginal zone (MZ) B cells, also

referred to as positive B cell selection[2–5]. Some immature B cells mature directly in the bone marrow[6,7], although this process is generally less studied.

V(D)J recombination inevitably creates the problem of self-reactivity. The clonal selection theory proposed that the period of antibody randomization is confined to a short time window during B cell development during which premature antigenic encounter results in elimination (clonal deletion) and self-tolerance instead of clonal expansion and antibody production[8]. Clonal deletion of immature B cells was first demonstrated in transgenic mice expressing a monoclonal self-reactive BCR[9]. Alternative tolerance mechanisms include silencing of the transgenic BCR by rearrangement of endogenous light chains (receptor editing)[10,11], and anergy[12]. Based on whether self-

[1]Experimental Immunology Branch, Center for Cancer Research, National Cancer Institute, National Institutes of Health, Bethesda, MD, USA. [2]Flow Cytometry Section, Research Technologies Branch, National Institute of Allergy and Infectious Diseases, National Institutes of Health, Bethesda, MD, USA. [3]Key Laboratory of Medical Molecular Virology (MOE/NHC/CAMS), Shanghai Institute of Infectious Disease and Biosecurity, School of Basic Medical Sciences, Fudan University, Shanghai, China. [4]These authors contributed equally: Mikala JoAnn Simpson, Anna Minh Newen. ✉e-mail: christian.mayer@nih.gov

antigen is first encountered in the bone marrow or in the periphery, B cell tolerance mechanisms are categorized as central and peripheral, respectively. Subsequent work suggested that receptor editing rather than clonal deletion is the main mechanism of central B cell tolerance[13,14]. To what extent clonal deletion contributes to peripheral tolerance is not entirely clear.

Up to 97% of newly developing immature B cells were suggested to undergo apoptosis prior to becoming mature B cells[15,16]. If clonal deletion is rare during central tolerance, this could only be explained by either massive clonal deletion during peripheral tolerance or by high levels of immature B cell apoptosis in the bone marrow and/or periphery by mechanisms other than clonal deletion. The anatomic sites and mechanisms of apoptosis during physiologic B cell development are therefore incompletely understood.

We previously developed Rosa26[INDIA] mice in which all B cells express a genetically encoded indicator of apoptosis (INDIA)[17]. INDIA consists of the FRET pair mNeonGreen and mRuby2. The linker region contains the active caspase-3 (aCasp3) target sequence DEVDG. Thus, there is a default FRET signal in aCasp3[neg] viable cells. Once aCasp3 is generated during the early phase of apoptosis it cleaves INDIA resulting in a loss of FRET[17]. Rosa26[INDIA] mice offer several advantages over traditional apoptosis detection methods including fluorescent caspase substrates, anti-aCasp3 antibodies, terminal deoxynucleotidyl transferase dUTP nick end labeling (TUNEL), and Annexin-V staining. These include the sensitive detection of aCasp3 rather than downstream events without a requirement for fixation or intracellular staining.

In this work, we employ Rosa26[INDIA] mice to gain insights into apoptosis during normal B cell development and its role in promoting self-tolerance. Our findings suggest that minimal B cell apoptosis occurs in the bone marrow but that mechanisms other than clonal deletion dominate central B cell tolerance. In contrast, transitional B cells undergo high levels of apoptosis in the periphery as soon as they enter the circulation from bone marrow sinusoids. However, most of this aCasp3-dependent transitional B cell apoptosis appears to reflect a lack of survival signaling rather than clonal deletion. This improves our understanding of self-tolerance and of primary B cell repertoire selection by aCasp3-dependent apoptosis.

## Results

### Quantitation of apoptosis during physiologic B cell development

To determine when apoptosis occurs during physiologic B cell development, we analyzed Rosa26[INDIA] mice by flow cytometry and quantitated the percentage of B cells harboring the FRET[neg] fluorescent indicator state which depends on cleavage by aCasp3 (Fig. 1a)[17]. Rosa26[INDIA] mice sensitively detected the vast majority of aCasp3[+] B cells in bone marrow and spleen (Supplementary Fig. 1a, b). In the bone marrow, apoptosis was generally low but significantly elevated in Fr. D pre-B cells ($0.288 \pm 0.030\%$ FRET[neg], $p < 0.0001$) and Fr. E immature B cells ($0.167 \pm 0.012\%$ FRET[neg], $p = 0.0003$) compared with Fr. F mature recirculating B cells ($0.090 \pm 0.007\%$ FRET[neg]) (Fig. 1b, e and Table 1). Apoptosis of earlier B cell precursors (Fr. B/C) was slightly lower compared with Fr. F. In contrast to the low levels of B cell apoptosis in the bone marrow, transitional B cells underwent high levels of apoptosis in the blood and spleen (average 1.034% to 2.025% FRET[neg]; $p < 0.0001$) compared with mature follicular (FO) B cells at these sites (average 0.112% to 0.132 % FRET[neg]; Fig. 1c–e and Table 1). When resolving T1 and T2 transitional B cell stages we found that apoptosis peaked in T1 cells in both blood and spleen (average 1.768% to 2.755 % FRET[neg]), and gradually declined from T2 cells (average 0.426% to 0.867% FRET[neg]) to FO B cells (Fig. 1f and Table 1). Anergic (T3) B cells have a shorter life span than FO B cells[18,19]. Indeed, blood and spleen T3 cells contained significantly more FRET[neg] cells (average 0.265% to 0.284% FRET[neg]) than FO B cells (Fig. 1f and Table 1). Less splenic MZ than FO B cells were FRET[neg] ($0.044 \pm 0.005\%$, $p < 0.0001$; Fig. 1d, e), consistent with increased MZ B cell longevity[20].

In vivo treatment with dexamethasone to acutely induce apoptosis resulted in an 8 to 98-fold increase of FRET[neg] cells for all B cell subsets and tissues examined (Supplementary Fig. 1f–h). To further validate Rosa26[INDIA] mice, we also induced apoptosis in vitro by culturing isolated cell suspensions for 1 h at 37 °C, or at 4 °C as control, and simultaneously quantitated FRET[neg] cells and intracellular aCasp3 in the same samples (Supplementary Fig. 1i–k). Under 4 °C control conditions, we obtained similar frequencies of FRET[neg] cells as in previous experiments, while 37 °C culture increased FRET[neg] cells dramatically across all B cell subsets, comparable to in vivo treatment with

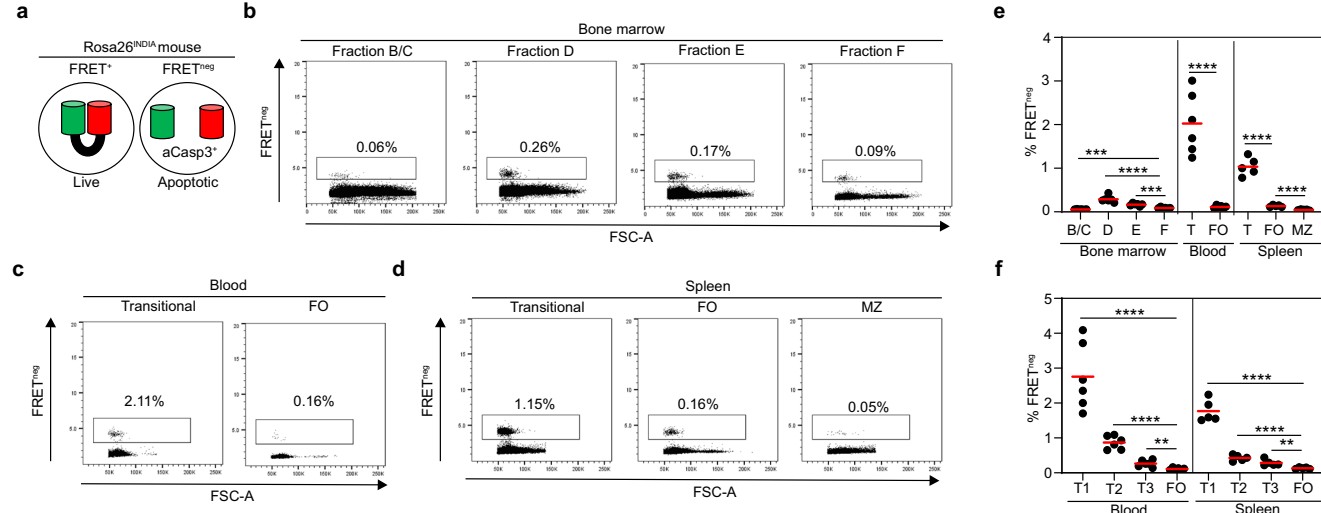

**Fig. 1 | Quantitation of apoptosis during physiologic B cell development.** Rosa26[INDIA] mice were analyzed by flow cytometry (red bars: mean values, B/C: pro/pre-B cells, D: small pre-B cells, E: immature B cells, F: mature recirculating B cells, T: transitional B cells, T1: transitional 1 B cells, T2: transitional 2 B cells, T3: anergic B cells, FO: mature follicular B cells, MZ: marginal zone B cells). Gating was done as shown in Supplementary Fig. 1c–e. **a** Schematics of Rosa26[INDIA] apoptosis indicator mice. **b**–**d** Representative dot plots depict forward scatter (FSC-A) and FRET loss of indicated B cell developmental stages in **b** bone marrow, **c** blood and **d** spleen. Experiments were repeated three time independently with similar results. **e, f** Quantitation of apoptotic FRET[neg] cells. Data are combined from three independent experiments ($n = 6$ mice for bone marrow and blood, $n = 5$ mice for spleen). **e** ****$p < 0.0001$, B/C vs F ***$p = 0.0009$, E vs F ***$p = 0.0003$. **f** ****$p < 0.0001$, Blood **$p = 0.0035$, Spleen **$p = 0.0069$; unpaired two-tailed t-test. Source data are provided as a Source Data file.

**Table 1 | Frequencies of FRETneg B cell populations in Rosa26INDIA mice**

| B cell population | % FRETneg (Mean ± SEM) |
|---|---|
| **Bone marrow** | |
| Pro-B (Fraction B/C) | 0.057 ± 0.002 |
| Small pre-B (Fraction D) | 0.288 ± 0.030 |
| Immature (Fraction E) | 0.167 ± 0.012 |
| Mature recirculating (Fraction F) | 0.090 ± 0.007 |
| **Blood** | |
| Transitional total | 2.025 ± 0.286 |
| Transitional 1 (T1) | 2.755 ± 0.390 |
| Transitional 2 (T2) | 0.867 ± 0.078 |
| Anergic (T3) | 0.265 ± 0.038 |
| Follicular (FO) | 0.112 ± 0.013 |
| **Spleen** | |
| Transitional total | 1.034 ± 0.093 |
| Transitional 1 (T1) | 1.768 ± 0.142 |
| Transitional 2 (T2) | 0.426 ± 0.038 |
| Anergic (T3) | 0.284 ± 0.041 |
| Follicular (FO) | 0.132 ± 0.010 |
| Marginal zone (MZ) | 0.044 ± 0.005 |

dexamethasone (Supplementary Fig. 1g and j). Intracellular aCasp3-staining was low in all bone marrow B cell subsets but clearly increased in splenic T1 and T2 cells compared with FO B cells under 4 °C control conditions (Supplementary Fig. 1k). Notably, intracellular aCasp3-staining was not sensitive enough to reveal differences in B cell subsets exhibiting low levels of apoptosis including bone marrow B cell subsets, splenic FO, and splenic MZ B cells, which was easily possible by quantitating FRETneg cells in the same samples even after fixation (Supplementary Fig. 1j, k). Culture at 37 °C increased the percentages of aCasp3+ cells across all B cell subsets and to levels comparable to the percentages of FRETneg cells (Supplementary Fig. 1j, k). We conclude that Rosa26INDIA mice are a sensitive indicator of aCasp3-mediated apoptosis, and that physiologic murine B cell development is associated with relatively low percentages of FRETneg cells in the bone marrow and with high percentages of FRETneg T1 B cells in the periphery.

**Quantitation of B cell death by FLICA-VAD and Annexin-V staining**

Strasser, Takeguchi and Wright et al. have independently shown that defective mitochondrial apoptosis results in increased numbers of several B cell subsets[21–23]. Additionally, defective death receptor-mediated apoptosis, for example downstream of Fas, results in lymphoproliferation[24]. No such effects on B cells have been reported in mice with defective aCasp3-independent cell death (e.g. pyroptosis and necroptosis). It is therefore quite well established that most steady state B cell apoptosis requires either the mitochondrial apoptosis pathway or the death receptor apoptosis pathway both of which require aCasp3.

Nevertheless, for comparison and to explore whether aCasp3-independent cell death might operate during B cell development, we next used a FLICA-VAD polycaspase probe that detects all active caspases. Cells from untreated Rosa26INDIA mice stained at 4 °C (control) had approximately 3 to 26 times more FLICA-VAD+ than FRETneg and aCasp3+ B cells except for splenic T1 cells that were comparable (Supplementary Fig. 1k and Supplementary Fig. 2a, b; Table 2). The percentages of FLICA-VAD+ and FRETneg cells after dexamethasone treatment were more comparable (Supplementary Fig. 2a, b; Table 2). When performing FLICA-VAD staining at 37 °C which is the manufacturer's recommended protocol, there were also 3 to 21 times more

**Table 2 | Comparison of FRETneg, FLICA-VAD+ and Annexin-V+ B cell populations in Rosa26INDIA mice**

| Condition | Measurement | Bone marrow | | | | Spleen | | | |
|---|---|---|---|---|---|---|---|---|---|
| | | Pro-B (Fr. B/C) | Small pre-B (Fr. D) | Immature (Fr. E) | Mature recirculating (Fr. F) | Transitional 1 (T1) | Transitional 2 (T2) | Follicular (FO) | Marginal zone (MZ) |
| Untreated 4 °C | % FRETneg ± SEM | 0.163 ± 0.015 | 0.338 ± 0.024 | 0.165 ± 0.013 | 0.089 ± 0.016 | 1.590 ± 0.229 | 0.250 ± 0.047 | 0.118 ± 0.026 | 0.066 ± 0.011 |
| Untreated 4 °C | % FLICA-VAD+ ± SEM | 1.718 ± 0.064 | 1.138 ± 0.033 | 1.910 ± 0.205 | 1.290 ± 0.124 | 1.713 ± 0.158 | 1.728 ± 0.154 | 0.900 ± 0.066 | 1.703 ± 0.179 |
| Untreated 4 °C | % Annexin-V+ ± SEM | 9.473 ± 1.917 | 4.290 ± 0.328 | 3.928 ± 0.191 | 2.438 ± 0.184 | 6.920 ± 0.530 | 6.995 ± 0.494 | 4.703 ± 0.399 | 4.598 ± 0.789 |
| Dexa 4 °C | % FRETneg ± SEM | 1.233 ± 0.229 | 2.718 ± 0.387 | 1.915 ± 0.191 | 1.685 ± 0.349 | 7.493 ± 1.049 | 2.623 ± 0.435 | 1.610 ± 0.395 | 2.523 ± 0.453 |
| Dexa 4 °C | % FLICA-VAD+ ± SEM | 2.618 ± 0.347 | 2.025 ± 0.204 | 2.425 ± 0.314 | 2.025 ± 0.343 | 6.013 ± 0.631 | 3.788 ± 0.481 | 2.408 ± 0.386 | 2.830 ± 0.321 |
| Dexa 4 °C | % Annexin-V+ ± SEM | 12.140 ± 1.287 | 5.105 ± 0.592 | 4.083 ± 0.652 | 3.128 ± 0.919 | 10.190 ± 1.340 | 6.460 ± 0.768 | 5.238 ± 0.879 | 5.130 ± 1.435 |
| Untreated 37 °C | % FRETneg ± SEM | 1.290 ± 0.032 | 3.745 ± 0.326 | 2.985 ± 0.507 | 2.363 ± 0.504 | 12.480 ± 2.249 | 2.468 ± 0.480 | 1.270 ± 0.196 | 1.620 ± 0.364 |
| Untreated 37 °C | % FLICA-VAD+ ± SEM | 8.158 ± 1.227 | 10.100 ± 0.518 | 10.660 ± 0.988 | 10.000 ± 1.501 | 18.090 ± 3.129 | 14.940 ± 2.685 | 7.223 ± 1.165 | 23.650 ± 4.061 |

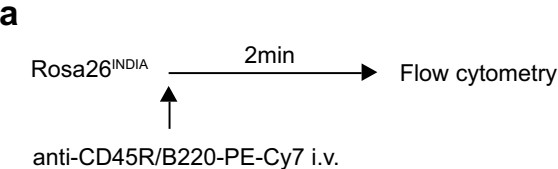

**a**

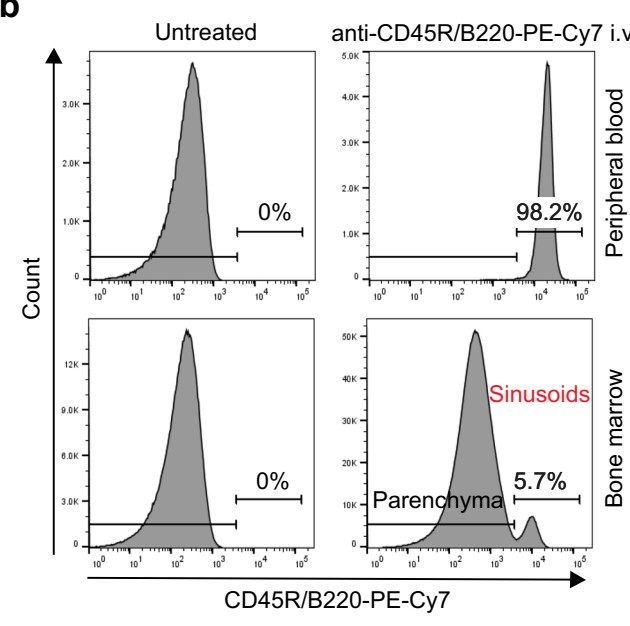

**b**

**c**

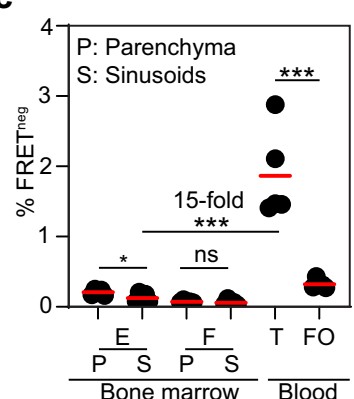

**Fig. 2 | Immature B cell apoptosis upon release from bone marrow sinusoids into the blood circulation.** Rosa26[INDIA] mice were injected intravenously with anti-CD45R/B220-PE-Cy7 2 min prior to euthanasia. Untreated animals served as controls. Gating of bone marrow and blood B cells was performed as in Supplementary Fig. 1c except intravascular anti-CD45R/B220-PE-Cy7 was used to label bone marrow sinusoids, CD19 was used to stain B cells instead of CD45R/B220, and dead cells were excluded using Zombie NIR Fixable Viability Kit instead of DAPI. **a** Schematics of the bone marrow sinusoid labeling experiment. **b** Representative histograms show PE-Cy7 labeling of CD19[+]Lineage(CD4.CD8α, NK1.1, F4/80, Ly-6G, Ter119)[neg]Zombie-NIR[neg]mRuby2[+] B cells in either peripheral blood or bone marrow. All B cells in blood are labeled, as expected. Labeling in the bone marrow marks B cells in sinusoids (red text), unlabeled B cell reside in the parenchyma (black text). Non-injected mice show no labeling in either blood or bone marrow. The experiment was repeated three times independently with similar results. **c** Quantitation of FRET[neg] cells among Fr. E immature B cells and Fr. F mature recirculating B cells in the bone marrow parenchyma (P: non-labeled) or sinusoids (S: labeled). Labeled transitional (T) and mature follicular (FO) B cells in blood are shown as controls. Results are combined from three independent experiments with similar results (*n* = 5 mice). E bone marrow sinusoids vs T blood ***p = 0.0003, T blood vs FO blood ***p = 0.0006, *p = 0.042, not statistically significant (ns) p = 0.51; unpaired two-tailed t-test. Source data are provided as a Source Data file.

FLICA-VAD[+] than FRET[neg] and aCasp3[+] B cells under the same conditions except for splenic T1 cells that were comparable (Supplementary Fig. 1k, 2a, b; Table 2). These results could point towards caspase activity other than aCasp3 in most B cell subsets but without any marked regulation during B cell development.

We also performed Annexin-V staining which detects phosphatidylserine exposure during apoptosis and other types of cell death. Interestingly, Annexin-V[+] cells were infrequent in mRuby2[+]DAPI[neg] B cells but were highly increased in mRuby2[neg]DAPI[neg] B cells in both bone marrow and spleen of Rosa26[INDIA] mice (Supplementary Fig. 2c, d). DAPI[+] B cells that have lost membrane integrity served as positive control (Supplementary Fig. 2c, d). Comparison of FSC-A and SSC-A profiles among mRuby2[+]DAPI[neg], mRuby2[+]DAPI[neg]FRET[neg], mRuby2[+]DAPI[neg]Annexin-V[+] and DAPI[+]Annexin-V[+] B cells revealed that mRuby2[+]DAPI[neg]FRET[neg] cells more closely resembled viable cells whereas mRuby2[+]DAPI[neg]Annexin-V[+] closely resembled dead cells (Supplementary Fig. 2e–g).

Based on these findings, we did not gate mRuby2[+] cells when comparing Annexin-V staining during B cell development. In untreated

Rosa26[INDIA] mice there were 4 to 70 times more Annexin-V[+] than FRET[neg] and aCasp3[+] cells across all B cell subsets (Supplementary Fig. 1j, k, 2h; Table 2). In the bone marrow, all B cell subsets had increased percentages of Annexin-V[+] cells compared with mature recirculating B cells with highest Annexin-V staining in Fraction B/C (Supplementary Fig. 2h). Annexin-V staining was also higher in splenic transitional compared with mature B cells. In contrast to our analyses of FLICA-VAD[+] and FRET[neg] cells, Annexin-V[+] cells were not substantially increased in any B cell population after dexamethasone treatment (Supplementary Fig. 2h). Nevertheless, the percentages of FLICA-VAD[+], FRET[neg] and Annexin-V[+] cells after dexamethasone treatment differed less than 3-fold across all B cell subsets except for Fr. B/C cells that contained 5 to 10 times more Annexin-V+ cells (Table 2).

In summary, analysis of FRET[neg] cells in Rosa26[INDIA] mice captures early apoptotic aCasp3[+] B cells, whereas Annexin-V staining marks B cells in a later stage of apoptosis or other forms of cell death. B cell apoptosis requires aCasp3 and the results and conclusions of the present study are focused on detecting aCasp3-dependent apoptosis. It will be interesting to investigate aCasp3-independent cell death in future studies to rule out or establish a contribution to B cell development.

### Immature B cell apoptosis upon release from bone marrow sinusoids

We next distinguished immature B cells in the bone marrow parenchyma from those retained in bone marrow sinusoids prior to release into the blood circulation by intravascular labeling of Rosa26[INDIA] mice[25] (Fig. 2a, b). FRET[neg] cells were quantitated in bone marrow parenchyma, sinusoids, and blood. Immature B cells in both bone marrow sinusoids and parenchyma contained a low fraction of FRET[neg] cells while blood transitional B cells had above tenfold more FRET[neg] cells (Fig. 2c). Thus, the release of immature B cells from bone marrow sinusoids into the blood circulation is associated with a dramatic increase in apoptosis.

### Predominant B cell apoptosis in the periphery

In order to examine the anatomic sites of B cell apoptosis in Rosa26[INDIA] mice further, we first determined total cell numbers for each B cell subset (Fig. 3a), which were consistent with published data for C57Bl/6 mice[16,26]. We assumed blood contains 0.2 million total transitional B cells and 0.8 million FO B cells (2 ml total blood volume, 5000 leukocytes/μl, 10% B cells of which 20% are transitional B cells and 80% are FO B cells). Additional assumptions are: (1) FRET[neg] B cells persist for

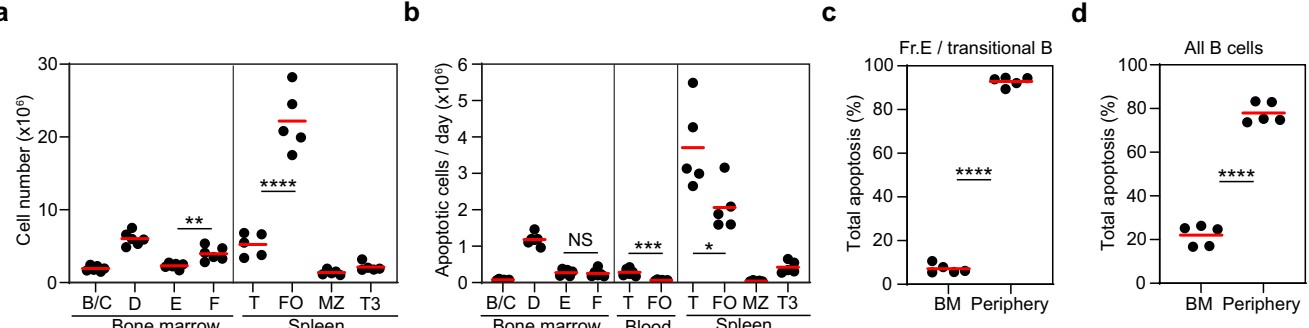

**Fig. 3 | Predominant B cell apoptosis in the periphery.** Rosa26^INDIA mice were analyzed by flow cytometry (red bars: mean values, B/C: pro/pre-B cells, D: small pre-B cells, E: immature B cells, F: mature recirculating B cells, T: transitional B cells, T3: anergic B cells, FO: mature follicular B cells, MZ: marginal zone B cells). Gating is shown in Supplementary Fig. 1c–e. **a** Total numbers of B cell subsets in bone marrow (2 femurs + 2 tibiae) and spleen calculated from total live cell counts (Neubauer chamber) and from the fraction of each B cell subset among live cells determined by flow cytometry. **b** Estimated number of B cell subsets undergoing apoptosis per day. **c** Fraction of all apoptotic Fr. E/transitional B cells from **b** undergoing apoptosis in the bone marrow or the periphery (blood and spleen). **d** Same as **c** but for all B cell subsets. **a–d** Data are combined from three independent experiments with similar results (**a, b** n = 6 mice for bone marrow and blood, n = 5 mice for spleen; **c, d**: n = 5 mice). **a–d** ****p < 0.0001, **a** **p = 0.0027, **b** ***p = 0.0003, *p = 0.025, not statistically significant (ns) p = 0.73; unpaired two-tailed t-test). Source data are provided as a Source Data file.

20.6 minutes until they disintegrate into small apoptotic bodies which are excluded from the flow cytometry gates used to detect FRET^neg cells (this is based on intravital microscopy of FRET^neg GC B cells[17]); (2) this apoptotic process is conserved for all B cells; (3) all FRET^neg B cells have identical clearance rates; (4) the lifetime of FRET^neg cells is much shorter than the lifetime of any B cell subset examined[16,17]. We then calculated the absolute numbers of each B cell subset undergoing apoptosis per day based on the measured total cell numbers, the measured fractions of FRET^neg cells and their assumed loss every 20.6 min. According to these estimates, on average 0.27 million immature bone marrow B cells, 0.28 million blood transitional B cells, and 3.71 million spleen transitional B cells undergo apoptosis per day (Fig. 3b). These data suggest that above 90% of immature/transitional B cell apoptosis occurs in the periphery (blood and spleen) and less than 10% in the bone marrow (Fig. 3c). When considering all B cell subsets combined, approximately 80 % of apoptosis occurs in the periphery and 20% in the bone marrow (Fig. 3d). Summarizing, both the directly measured fraction of FRET^neg B cells and their estimated total numbers in Rosa26^INDIA mice suggest that the periphery rather than the bone marrow is a major site of immature/transitional B cell apoptosis.

**Limited clonal deletion during physiologic B cell development**
Developing B cells might undergo apoptosis when receiving strong BCR signaling after binding multivalent self-antigens, resulting in clonal deletion[9,27]. To test this possibility, we isolated single FRET^+ and FRET^neg Rosa26^INDIA B cells, sequenced and cloned their BCR genes (Fig. 4a). Autoreactive and polyreactive BCRs are enriched with positively charged amino acids in the IgH complementarity-determining region 3 (CDR3) and with long IgH CDR3s[28,29]. The proportions of FRET^+ B cells with ≥ 3 positive IgH CDR3 charges only modestly decreased during development (Supplementary Fig. 3a). FRET^neg early immature, immature, and transitional B cell IgH CDR3 charges were not significantly different to FRET^+ cells, yet more B cells with four positive charges were seen which were rare in any FRET^+ B cell compartment (Supplementary Fig. 3a). The mean IgH CDR3 length did not notably change during B cell development, nor was there a significant difference in viable and apoptotic B cells (Supplementary Fig. 3b).

Since sequence features cannot accurately predict BCR binding properties, we next expressed 248 monoclonal antibodies (mAbs) corresponding to the BCRs of single FRET^+ and FRET^neg B cells across development and tested polyreactivity and autoreactivity (Fig. 4a, Supplementary Data 1). About 20% of FRET^+ early immature bone marrow B cells were polyreactive, which dropped to 5%–6% in FRET^+ immature B cells and splenic transitional B cells (Fig. 4b and Supplementary Fig. 3c). Polyreactivity further diminished in FRET^+ FO B cells. Compared with FRET^+ counterparts, FRET^neg B cells were not significantly enriched in polyreactivity in any compartment tested (Fig. 4b and Supplementary Fig. 3c, d). Five percent of FRET^+ early immature and immature B cells displayed strong polyreactivity, while only weakly polyreactive BCRs remained at the transitional B cell stage (Fig. 4b, c and Supplementary Fig. 3c, d). Again, apoptotic FRET^neg early immature and immature B cells did not contain significantly more polyreactive BCRs (Fig. 4b, c and Supplementary Fig. 3c, d). Modest enrichment in polyreactivity to five antigens—yet also not significant—was only seen at the immature B cell stages in the bone marrow.

We next developed a flow cytometry-based screening assay coupled with confocal microscopy or imaging flow cytometry (FLOW-MIST) to identify nuclear/subnuclear, cytoplasmic and/or membrane autoantibodies using primary mouse splenocytes as target cells (Fig. 4d). FLOWMIST sensitively and accurately detected autoreactive control antibodies previously tested by indirect immunofluorescence of HEp-2 cells (Fig. 4e)[28,30]. After screening the 248 mAbs by FLOWMIST, we identified several autoreactive antibodies recognizing cytoplasmic and/or nuclear antigens (Supplementary Fig. 3e, f). Some of these included highly polyreactive BCRs (see Supplementary Data 1). Autoreactivity gradually declined from 13% in FRET^+ early immature B cells to less than 1/32 ( < 3%) in splenic transitional B cells but increased again to 10% in mature FO B cells (Fig. 4f), consistent with the finding that weak self-reactivity can promote positive selection into the mature FO B cell compartment[31,32]. Again, apoptosis had no detectable role at eliminating autoreactive B cells throughout B cell development (Fig. 4f and Supplementary Fig. 3e), similar to results for GC B cells[17,28].

Finally, we performed immunoblot assays of all 248 antibodies on C57Bl/6 J spleen, bone marrow, kidney, and thyroid lysates. All antibodies that consistently reacted to one or more tissue lysates had already been identified as either polyreactive and/or autoreactive (Supplementary Fig. 3g, h and Supplementary Data 1), suggesting that no autoantigens in the bone marrow or the peripheral tissues tested were missed by performing polyreactivity and FLOWMIST assays.

Summarizing, we detected a significant reduction of self-reactivity/polyreactivity from early immature to immature FRET^+ B cells in the bone marrow (Fig. 4g and Supplementary Fig. 3i), consistent with findings in humans and indicating frequent BCR engagement with self-antigens at this site[29]. However, these B cell compartments contained low frequencies of FRET^neg cells and the BCRs of these apoptotic

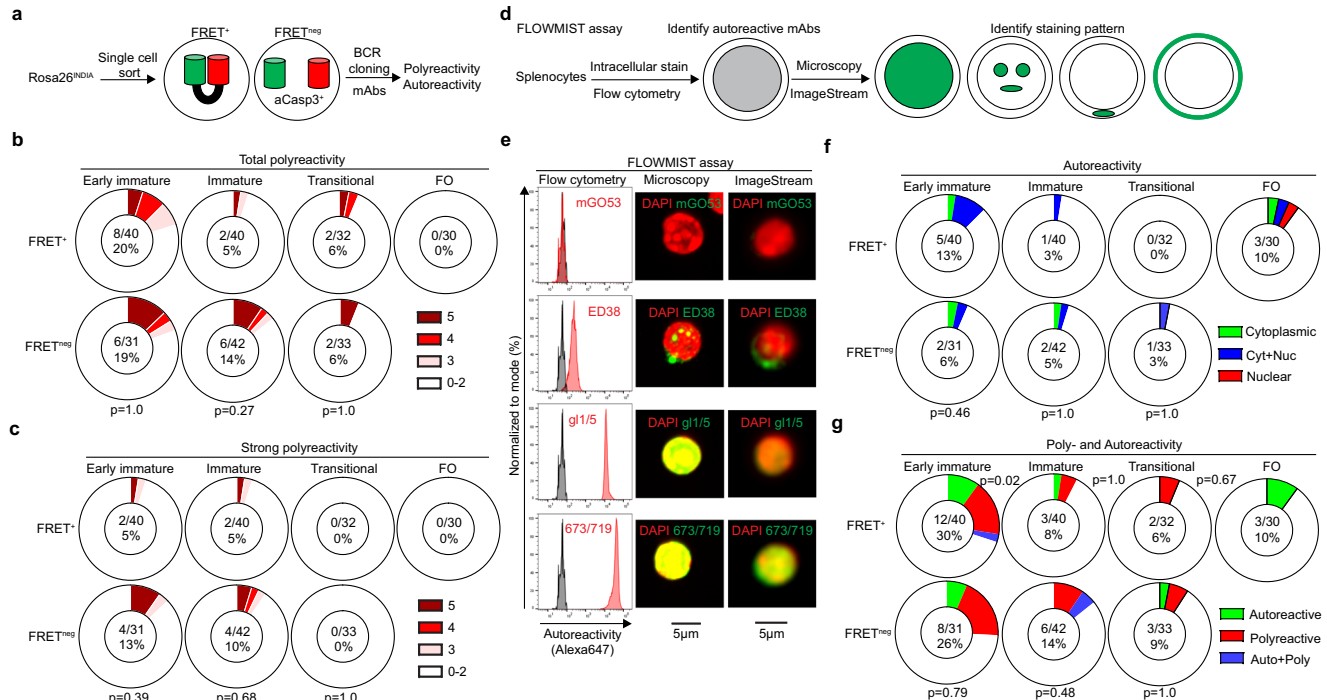

**Fig. 4 | Limited clonal deletion of self-reactive B cells during physiologic B cell development.** Live (FRET⁺) and apoptotic (FRETⁿᵉᵍ) Rosa26^INDIA B cell compartments were single-cell FACS-sorted. Gating was done as described in Supplementary Fig. 1c, e with minor changes as descried in the methods. Monoclonal antibodies (mAbs) are derived from two to three independent experiments. Ig genes were cloned, expressed as mAbs, and tested. Pie chart centers show the number of reactive and total mAbs tested, and the percentage of reactive mAbs. **a** Experimental outline. **b, c** Summary of polyreactivity determined by ELISA showing **b** total or **c** strong polyreactivity. Red intensity depicts polyreactivity to 3, 4 or 5 antigens, white represents reactivity to two or less antigens. All mAbs were tested in parallel once against each antigen representing five independent experiments. Polyreactivity was confirmed using dot blot as an independent method. **d–f** mAb autoreactivity testing by the flow cytometry-based screening assay coupled with confocal microscopy or imaging flow cytometry (FLOWMIST) assay. **d** Experimental outline. **e** FLOWMIST

analysis of control mAbs mGO53 (non-reactive), ED38 (cytoplasm/nucleus-reactive), 673/719 and its germline-revertant gl1/5 (both anti-nuclear). Left, flow cytometry histograms (unstained cells represented in gray). Right, fluorescent staining pattern (green) determined by confocal microscopy and ImageStream. DAPI is shown in red. Scale bars are 5 μm. Cropped micrographs depict single cells in the center. Experiments were performed two times independently with similar results. **f** Summary of autoreactivity among indicated B cells compartments. Green (anti-cytoplasmic), blue (anti-cytoplasmic and nuclear), red (anti-nuclear). **g** Summary of autoreactivity (green) and polyreactivity (red) among indicated B cell compartments. Blue depicts antibodies that are autoreactive and polyreactive. **a–g** The p values for comparing FRET⁺ and FRETⁿᵉᵍ cells are shown below the pie charts, the p values for comparing FRET⁺ B cell subsets are shown in between the pie charts (two-tailed Fisher's exact test). Reactive mAbs were confirmed with two independent experiments or two independent methods with similar results.

immature B cells displayed comparable self-reactivity than their FRET⁺ counterparts (Fig. 4g, Supplementary Fig. 3i). Transitional B cells in the spleen included a high percentage of FRETⁿᵉᵍ cells but above 90% of BCRs from these apoptotic transitional B cells were not self-reactive (Fig. 4g, Supplemental Fig. 3i). Additionally, no significant reduction in self-reactivity was seen from transitional to mature B cells, suggesting overall less frequent and/or weaker BCR engagement with self-antigens in the periphery than in the bone marrow. Based on these data we suggest that most early removal of self-reactivity in the bone marrow is not mediated by aCasp3-dependent clonal deletion but by another mechanism, most likely receptor editing[13,14]. Additionally, the high level of apoptosis observed in peripheral transitional B cells is unexplained by binding to self-antigens. Thus, clonal deletion appears to make a minor contribution to the apoptosis detectable during physiologic B cell development.

## Mechanism of peripheral transitional B cell apoptosis

If most peripheral transitional B cells are not signaled to undergo apoptosis by binding self-antigens (Fig. 4), another possibility is that they are not signaled to survive. Indeed, interconnected signals transmitted through the BCR and BAFF-R are required for splenic T1 B cell development into T2 and mature B cells, a process also referred to as positive B cell selection[3,4,33–35]. We thus hypothesized that many T1 B cells fail to receive positive selection signals and die by neglect. This would predict that T1 cells with lower IgM BCR expression receive less

BCR-dependent positive selection signals and undergo apoptosis preferentially.

To this end, we analyzed B cell development in Nur77^GFP mice which report recent BCR signaling in splenic transitional and mature B cells by GFP expression[36]. We found a small fraction of IgM^lo GFPⁿᵉᵍ cells among T1 B cells (average 3.5%), which gradually declined in T2/3 B cells (average 2%) and mature FO B cells (average 1.2%) (Fig. 5a, b). Inhibition of B cell apoptosis in Nur77^GFP Eμ-Bcl2^tg mice led to a marked accumulation of IgM^lo GFPⁿᵉᵍ T1 B cells (average 23%, $p = 0.0003$, Fig. 5a, b), which could indicate that T1 B cells with low IgM expression and BCR signaling normally die rapidly. Consistently, analysis of Rosa26^INDIA mice revealed that IgM^lo T1 B cells displayed significantly increased percentages of apoptotic FRETⁿᵉᵍ cells compared with IgM^hi T1 B cells in both blood and spleen (Supplementary Fig. 4a–c). In contrast, identically gated IgM^hi and IgM^lo FO B cells had similar fractions of FRETⁿᵉᵍ cells (Supplementary Fig. 4a–c), suggesting that T1 B cells are uniquely sensitive to changes in IgM expression compared with FO B cells. Despite inhibition of B cell apoptosis in Nur77^GFP Eμ-Bcl2^tg mice, IgM^lo GFPⁿᵉᵍ cells were depleted from T2/3 B cells and mature FO B cells (average 2.9–3.4%), indicating that the BCR also controls T1 B cell differentiation independently from promoting survival (Fig. 5a, b)[3].

Nur77^GFP Eμ Bcl2^tg mice also accumulated IgM^lo GFP⁺ T1 B cells compared with Nur77^GFP mice (Fig. 5a, c) which could reflect self-reactive B cells that signaled and downregulated IgM[36]. These

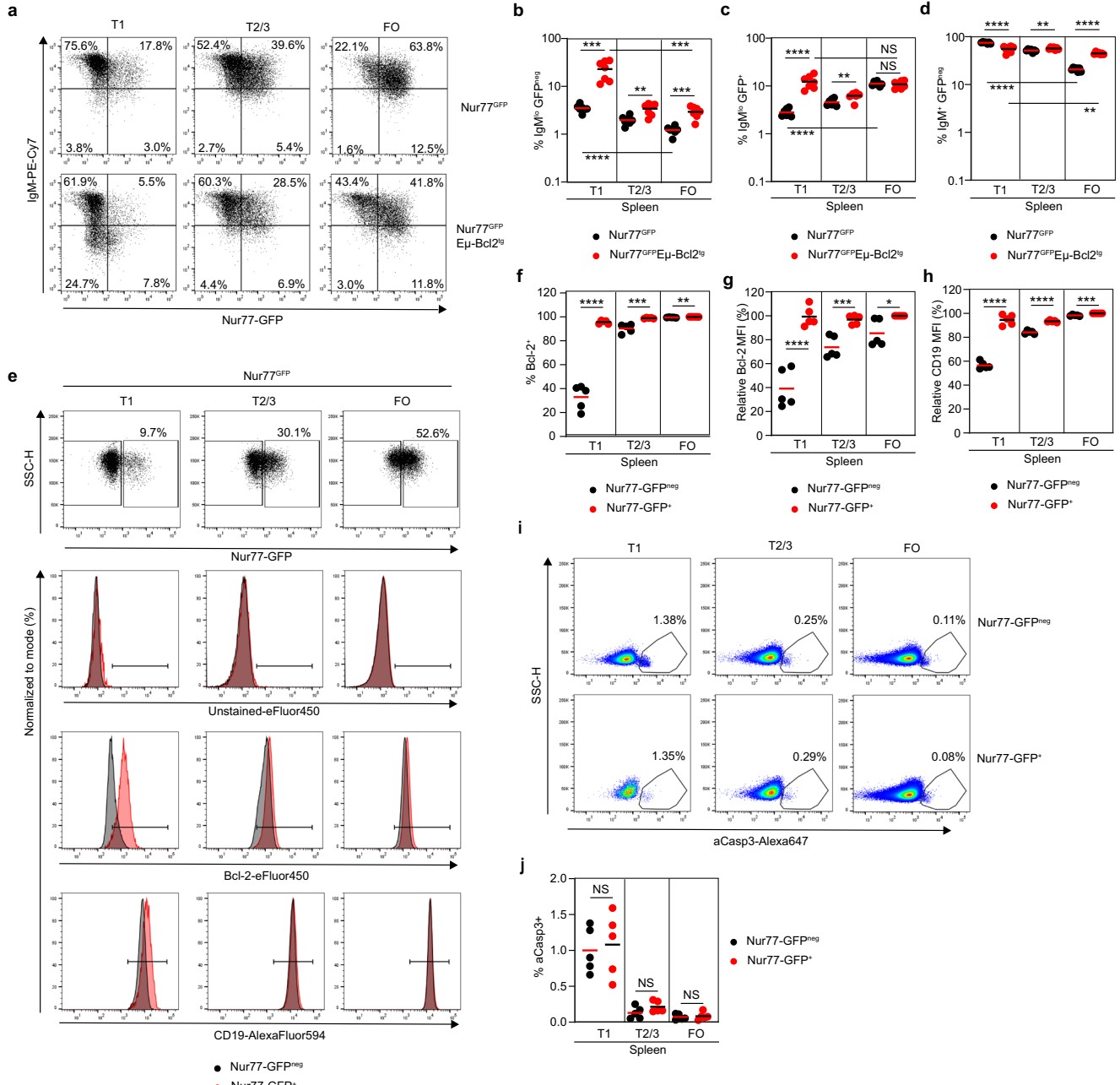

**Fig. 5 | Mechanism of peripheral transitional B cell apoptosis.** Spleens of Nur77GFP and Nur77GFPEμ-Bcl2tg mice were analyzed by flow cytometry (T1: transitional 1 B cells, T2/3: transitional 2/3 B cells, FO: mature follicular B cells, horizontal bars: mean values). Gating is shown in Supplementary Fig. 5. **a** Representative dot plots display GFP and IgM expression among indicated B cell subsets. The experiment was performed three times independently with similar results. **b–d** Quantitation of **b** IgMloGFPneg, or **c** IgMloGFP+, or **d** IgM+GFPneg B cells. Results are combined from three independent experiments (*n* = 7 mice). **b–d** Unpaired two-tailed t-test, ****$p$ < 0.0001. **b** T1 Nur77GFP vs T1 Nur77GFPEμ-Bcl2tg ***$p$ = 0.0003, T1 Nur77GFPEμ-Bcl2tg vs FO Nur77GFPEμ-Bcl2tg ***$p$ = 0.0002, FO Nur77GFP vs FO Nur77GFPEμ-Bcl2tg ***$p$ = 0.0001, **$p$ = 0.0025, **c** **$p$ = 0.0061, not statistically significant (ns) T1 Nur77GFPEμ-Bcl2tg vs FO Nur77GFPEμ-Bcl2tg $p$ = 0.3465, FO Nur77GFP vs FO Nur77GFPEμ-Bcl2tg $p$ = 0.6579, **d** T1 Nur77GFPEμ-Bcl2tg vs FO Nur77GFPEμ-Bcl2tg

**$p$ = 0.0064, T2/3 Nur77GFP vs T2/3 Nur77GFPEμ-Bcl2tg $p$ = 0.0087. **e–j** Spleen cells of Nur77GFP mice were stained intracellularly for Bcl-2 and active caspase-3 (aCasp3) and analyzed by flow cytometry. GFP+ and GFPneg subsets were compared. Results are from three independent experiments with similar results (*n* = 5 mice). **e** Representative dot plots show GFP expression and SSC-H. Histograms display intracellular staining for Bcl-2 and surface staining for CD19 (GFPneg: gray, GFP+: red). **f–h** Quantitation of **f** Bcl-2+ percentage or **g** Bcl-2 mean fluorescence intensity (MFI) or **h** CD19 MFI. MFIs are relative to GFP+ FO B cells. **f–h** Unpaired two-tailed t-test. ****$p$ < 0.0001, **f** ***$p$ = 0.0008, **$p$ = 0.0022, **g** ***$p$ = 0.0008, *$p$ = 0.0206, **h** ***$p$ = 0.0001. **i** Representative pseudocolor plots show aCasp3 vs side scatter (SSC-H). **j** Quantitation of aCasp3+ cells among indicated subsets and genotypes (unpaired two-tailed t-test). Not statistically significant (ns): T1 $p$ = 0.750, T2/3 $p$ = 0.138, FO $p$ = 0.729. Source data are provided as a Source Data file.

differences were lost during subsequent B cell development (Fig. 5a, c). Additionally, IgM+GFPneg B cells without recent BCR signaling gradually decreased during B cell development in Nur77GFP mice but accumulated upon Bcl-2 expression and persisted significantly in the FO B cell compartment of Nur77GFPEμ-Bcl2tg mice (Fig. 5a, d).

To directly assess the relationship between BCR signaling and T1 B cell survival, we intracellularly stained Nur77GFP B cells for Bcl-2 and aCasp3. While GFPneg T1 B cells were Bcl-2neg/lo, almost all GFP+ T1 B cells expressed high levels of Bcl-2 comparable to mature B cells (Fig. 5e–g). In contrast, most GFPneg T2/3 and FO B cells were Bcl-2+, and GFP+

counterparts only expressed slightly higher Bcl-2 levels (Fig. 5e–g). CD19, which is involved in BCR and BAFF-R signaling[37,38], was also most differentially expressed at the T1 B cell stage as a function of GFP expression (Fig. 5e, h).

GFP[neg] T1 B cells had a high level of aCasp3 staining (Fig. 5i, j), as expected (Fig. 1 and Supplementary Fig. 1k). Despite increased Bcl-2 and CD19 expression, GFP[+] T1 B cells were not rescued from apoptosis (Fig. 5i, j). These results suggest that T1 B cells are disadvantaged by apoptosis until they complete differentiation into T2 and mature B cells.

## Discussion

Large pre-B cells undergo massive proliferation in the bone marrow prior to forming quiescent small pre-B and immature B cells[1]. Assuming negligible effects by proliferation or circulation in the subsequent B cell developmental stages, this has been exploited to study B cell development by using pulse-chase labeling with BrdU which is incorporated into DNA during the S phase and can be detected using flow cytometry. The rate at which recently generated B cells enter each population was estimated from the linear portion of the BrdU labeling curve as 12 to 14 million for bone marrow immature B cells per day, 1.1 to 1.6 million for splenic transitional B cells per day, and 0.2 to 0.6 million for splenic mature B cells per day[15]. These and similar experiments were interpreted to reflect high levels of immature/transitional B cell loss by apoptosis, particularly in the bone marrow[15,16]. Immature B cell apoptosis in the bone marrow might be attributed to clonal deletion of self-reactive B cells, but several subsequent studies conflict with that hypothesis[13,14]. It therefore remained incompletely understood at which anatomic sites and developmental stages B cells undergo apoptosis during physiologic B cell development, and why.

We here filled some of these gaps by directly measuring aCasp3-dependent apoptosis across B cell development using Rosa26[INDIA] apoptosis indicator mice, and by estimating total numbers of apoptotic B cells. Our results suggest that most immature/transitional B cells undergo apoptosis in the periphery rather than the bone marrow. These findings are consistent with a suppressive effect of the bone marrow microenvironment on immature B cell apoptosis[39] and with infrequent clonal deletion at this site[13,14]. The factors negatively regulating B cell apoptosis in the bone marrow microenvironment are of interest for future investigation.

While every experimental system has advantages and limitations, the question arises why the results from our and other laboratories differ from the interpretations of BrdU pulse-chase experiments[15,16]. One interpretation in addition to cell loss by apoptosis is that not all BrdU-labeled immature B cells in the bone marrow flux to the BrdU-labeled splenic transitional B cell pool and are found at different sites. Indeed, approximately 50% of newly formed immature B cells were suggested to mature directly within the bone marrow[6]. Additionally, unexpected niches for B cell development were recently discovered in the meninges of the central nervous system[40]. Next, immature B cells undergoing receptor editing are developmentally arrested in the bone marrow[14]. Given that we find up to 30% self-reactive/polyreactive early immature B cells that can potentially undergo receptor editing, we predict that newly formed immature B cells leave the bone marrow at varying times. Finally, B cell loss could also occur in an aCasp3-independent manner. We found increased levels of polycaspase- and Annexin-V staining, but not aCasp3 staining, in Fr. B/C and immature B cells compared with mature recirculating B cells in the bone marrow. Whether this could reflect cCasp3-independent cell death is of interest for further investigation.

As one limitation we had to rely on the directly measured lifetime of FRET[neg] GC B cells[17] when estimating total numbers of B cells undergoing apoptosis per day. This lifetime could be different for other FRET[neg] B cells. On the other hand, apoptosis is a highly conserved biological process among different cell types and even among different species[41]. As another caveat we cannot exclude that the ability to detect FRET[neg] cells is counteracted by their clearance which could differ among various tissues. However, we show that immature B cells in two similar tissues (bone marrow sinusoids vs. peripheral blood circulation) show drastic differences in apoptosis. This makes it unlikely that these differences are solely due to differences in the clearance of apoptotic cells. Generally, this discussion is not specific to Rosa26[INDIA] mice but applies to every study of apoptotic cells that must rely on the fraction of apoptotic cells that can be detected with the methods used in each tissue under study. In fact, one strength of Rosa26[INDIA] mice compared with previously available methods is the early and sensitive detection of aCasp3[+] apoptotic cells without the need for fixation or intracellular staining.

Immature/transitional B cells that receive strong BCR signaling by self-antigens may undergo clonal deletion by apoptosis[9], as predicted by Burnet's clonal selection theory[8,42]. We here approached this problem in mice lacking BCR transgenes or transgenic BCR ligands. Our findings suggest that clonal deletion makes a minimal contribution to the total immature B cell apoptosis in bone marrow and spleen. This would predict that an aCasp3-independent mechanism removes most self-reactivity in the bone marrow, which is consistent with the conclusion that receptor editing is the main mechanism of central B cell tolerance[13,14]. Importantly, while apoptosis of autoreactive/polyreactive B cells appears to be a rare event during B cell development in normal mice, this does not preclude an important contribution to self-tolerance, as clearly demonstrated in BCR transgenic model systems[9,19,43].

While we cannot exclude that B cells reacting to self-antigens with such low affinity that they remain undetected in our assays are removed by apoptosis, this seems unlikely since apoptosis of developing B cells requires strong BCR stimulation by multivalent antigens[27]. Another possibility is that the cognate self-antigens were not present in our self-reactivity assays, which we also minimized by comprehensively testing antibody binding to primary mouse cells, to various mouse tissue lysates and to defined self-antigens by using several independent methods. Nevertheless, reactivity to peripheral tissues that were not tested cannot be excluded.

When comparing the amount of self-reactivity in FRET[+] B cell compartments in our study to other studies, one must consider that the mouse genetic background, environment, experimental models, methods, sensitivity, and cut-off for detecting autoreactivity can differ among laboratories and the results may not be directly comparable. However, our finding that 8%–30% of immature B cells in the bone marrow were autoreactive and/or polyreactive is consistent with the estimate that 25% of immature B cells undergo receptor editing, which is a known consequence of self-reactivity[44]. Moreover, our observation that most reduction in self-reactivity occurs from early immature to immature B cells in the bone marrow is similar to the findings by Nojima et al. who profiled the DNA reactivity of cultured single B cells from normal mice[45], and to findings in humans[29].

If peripheral transitional B cells are not frequently signaled to undergo apoptosis by recognizing self-antigens, then one alternative possibility is that they undergo apoptosis by default and aren't signaled to survive or differentiate into mature B cells. In support of this idea, BCR signaling components (e.g., Syk, CD79a, CD45, CD19) and BAFF/BAFF-R control transitional B cell survival and differentiation into mature B cells, a process also referred to as positive selection that shapes the mature B cell repertoire[2–5,46,47]. Importantly, such a mechanism would still ensure the removal of transitional B cells with high reactivity to self-antigens and could explain why these cells are apparently very rare in both the FRET[+] and FRET[neg] B cell repertoires. Removal could occur either at the T1 stage, T2 stage or after adopting an anergic T3 phenotype[18]. Consistent with the increased levels of T3 apoptosis and with increased T3 autoreactivity[18,45], inhibition of

apoptosis by deletion of Bcl2l11 can enhance survival and maturation of autoreactive T3 cells resulting in a loss of B cell tolerance[19].

We examined BCR signaling with the Nur77[GFP] reporter and found that only 10–20% of splenic T1 B cells exhibit recent BCR signaling. These cells expressed significantly higher levels of Bcl-2 and CD19 than their un-signaled counterparts but were not measurably protected from apoptosis. Two possible explanations are: (1) T1 cells need signals in addition to those generated by the BCR to induce survival; (2) GFP⁺ T1 cells are committed to survival but need to complete differentiation until the effect becomes measurable. Based on our data, we hypothesize that many peripheral transitional B cells fail positive selection and undergo death by neglect. A recent study by Hobeika and colleagues supports this idea because in vivo stimulation of T1 B cells by anti-IgM antibodies did not induce apoptosis as seen in vitro, but instead caused Bcl-2 expression, survival, and differentiation into T2 cells[48]. Transitional B cell apoptosis by lack of survival signaling and positive selection resembles the mechanism of B cell apoptosis in the light zones of GCs[17,49].

Transitional B cell positive selection warrants further investigation. One point to consider is that the BCR and BAFF-R receptor systems were shown to interact and even share signaling components[34,35,37]. It is therefore possible that the Nur77[GFP] reporter indicates processes in addition to antigen-triggered BCR signaling in vivo. Nevertheless, BCR-dependent positive selection predicts that weakly autoreactive T1 B cells have an advantage in positive selection, and this is what we and others observed[31,32]. Our single-cell sorting did not discriminate mature follicular B cells from anergic T3 cells which contain 10% to 30% autoreactive cells[18,45]. However, given that T3 cells make up about 7% of FO splenic B cells[18], we estimate that T3 cells could only have contributed 0.7% to 2.1% of autoreactivity to FO B cells, markedly below our measured number of 10%. Finally, positive selection may also depend on B cell-intrinsic properties such as tonic BCR signaling. Consistently, we found that T1 B cells with low IgM expression, for example due to suboptimal pairing of IgH/IgL, had significantly increased levels of apoptosis. Moreover, these IgM[lo] T1 cells expanded dramatically upon inhibition of apoptosis by expression of Bcl-2.

Open questions remain about the purpose of positive selection during peripheral B cell development at the expense of substantial cell loss by apoptosis. It is tempting to speculate that advantageous features are being selected and/or that detrimental properties are being avoided in the repertoire.

# Methods

## Mice and treatments

All animal procedures reported in this study that were performed by NCI-CCR affiliated staff were approved by the NCI Animal Care and Use Committee and in accordance with federal regulatory requirements, standards, and ethical regulations. All components of the intramural NIH ACU program are accredited by AAALAC International. Mice were maintained under specific pathogen-free conditions at NCI (Frederick and Bethesda) at a 12-h dark/light cycle, temperature range 68–79 °F, humidity range 30–70%. For bone marrow sinusoid B cell labeling, mice were injected intravenously with 2 µg PE-Cy7-conjugated anti-CD45R/B220 antibody (Clone RA3-6B2, Thermo Fisher Scientific, Catalog #25-0452-82) 2 min prior to euthanasia[25]. For some experiments, mice were injected with 0.5 mg dexamethasone intraperitoneally 4 hours prior to analysis. Rosa26[INDIA] mice[17] and Eµ-Bcl2[tg] mice[50] were both on a C57Bl/6 J background and a kind gift by Dr. Michel Nussenzweig (The Rockefeller University). Nur77[GFP] mice[36] were on a C57Bl/6 J background and a kind gift by Dr. Richard Hodes (NCI). C57Bl/6 J mice were purchased from Jackson Laboratories (Strain #664). The experiments in this study contain male and female mice and therefore findings do not apply to only one sex. Figures 1 and 3 used five male (bone marrow, blood) or four male (spleen) Rosa26[INDIA]

mice and one female Rosa26[INDIA] mouse (age 8.3 to 13.9 weeks). Figure 2 used four male Rosa26[INDIA] mice and one female Rosa26[INDIA] mouse (age 6.9 to 12.6 weeks). Single-cell sorting in Fig. 4 used two male and four female Rosa26[INDIA] mice (age 7.4 to 13.4 weeks). Figure 5a–d used zero male and seven female Nur77[GFP] mice (age 9.7 to 17.0 weeks), and four male and three female Nur77[GFP]Eµ-Bcl2[tg] mice (age 12.7 to 17.0 weeks). Figure 5e-j used three male and two female Nur77[GFP] mice (age 12.3 to 26.9 weeks).

## ELISA

Polyreactivity was assessed by measuring antibody binding to lipopolysaccharide (LPS, Sigma, L2637), Keyhole Limpet Hemocyanin (KLH, Sigma, H8283-100 MG), double stranded DNA (dsDNA, Sigma, D4522), single stranded DNA (ssDNA, prepared from dsDNA) and human insulin (Sigma, I9278) by ELISA[51]. Antigens were diluted in PBS (ssDNA, dsDNA, LPS, KLH: 10 µg/ml; Insulin: 5 µg/ml) and coated overnight at 4 °C using polystyrene high binding microplates (Corning, 9018). After blocking with PBS containing 2.5% (w/v) bovine serum albumin (BSA, Rockland, BSA-1000), 1 mM EDTA (Thermo Fisher, 15575-038) and 0.05% (v/v) Tween20 (Thermo Fisher, J20605-AP) for 2 h at room temperature, monoclonal antibodies and control antibodies (mGO53, non-reactive; ED38, highly polyreactive) containing human IgG1 constant regions were tested at 4 µg/ml and at three four-fold dilutions in PBS (binding for 2 h at room temperature) and were detected for 1 h at room temperature with peroxidase-conjugated goat anti-human IgG Fc (Jackson ImmunoResearch, 109-035-098, 1:1000 dilution). Peroxidase was revealed with 1-Step ABTS substrate solution (Thermo Fisher, 37615) for 30 min and absorbance at 405 nm was measured with a SpectraMax iD3 Multi-Mode Microplate Reader (Molecular Devices). $A_{405}$ values were corrected for PBS-only signal for each plate. Monoclonal antibodies binding to ≥ 3 different antigens at 4 µg/ml above a defined threshold were considered polyreactive. Those antibodies still binding to ≥ 3 different antigens at 1 µg/ml were considered strongly polyreactive.

## Flow cytometry and cell sorting

FACS buffer (PBS containing 0.1% w/v BSA and 2 mM EDTA) was used for cell isolation, incubation and acquisition if not stated otherwise. For Rosa26[INDIA] experiments, euthanized mice, buffers, and consumables contacting cells were pre-chilled and maintained on ice to minimize de novo apoptosis during cell isolation, staining and acquisition. All centrifugations and cell sorting were done at 4 °C.

Immediately after euthanasia peripheral blood was drawn from the inferior vena cava, mixed with 10 mM EDTA to prevent coagulation and added to 10 ml ACK lysis buffer (Quality Biological, 118-156-101). After incubation for 5 min on ice, cells were centrifuged for 7 min at 395 x g and washed with 10 ml of FACS buffer. Bone marrow was flushed out from femurs and tibiae. Single-cell suspensions were created from bone marrow and spleens by forcing the tissue through 70 µm cell strainers (Corning, 352350). For spleen, erythrocytes were lysed by resuspending pellets in 1 ml ACK lysis buffer (Quality Biological, 118-156-101) and incubating for 1 min on ice, followed by washing with FACS buffer. Blood leukocytes, bone marrow and spleen cell suspensions were then transferred to 96-well round bottom plates. All centrifugations were performed for 3 min at 395 × g.

For live cell staining, Fc receptors were first blocked for 15 min on ice with CD16/CD32 antibody (2.4G2, produced and purified by the EIB Flow Cytometry Core, 1:10 dilution). After centrifugation, cells were stained with antibodies to surface antigens (see Supplementary Table 1) for 45 min at 4 °C and washed three times with FACS buffer. If applicable, cells were then stained with APC-Fire750-conjugated streptavidin (BioLegend, 405250, 1:320 dilution) for 10 min at 4 °C and washed three times. For some experiments, cells were first stained with Far-Red fluorescent FLICA 660 VAD (Immunochemistry

Technologies, 9120) diluted in PBS for 1 h at 4 °C or at 37 °C prior to the staining steps above. For some experiments, cells were stained with AlexaFluor647-conjugated Annexin-V (Invitrogen, A23204, 1:20 dilution) in 1xAnnexin binding buffer (ThermoFisher, V13246) as the final step. Cells were resuspended in FACS buffer containing 0.2 µg/ml propidium iodide (PI) (Sigma-Aldrich, P4170) or 0.1 µg/ml DAPI (Sigma-Aldrich, D9542), or in 1xAnnexin binding buffer contain DAPI, prior to acquisition to exclude dead/necrotic cells.

For intracellular staining and for some live cell experiments where PI and DAPI were not compatible with the staining panel, cells were washed twice with PBS and stained using the Zombie NIR or Zombie UV Fixable Viability Kit (Biolegend, 423106 or 423108, diluted 1:500 in PBS) for 30 min at 4 °C prior to Fc blocking and staining of surface antigens. For intracellular staining, cells were then washed with PBS and incubated with Fixation/Permeabilization solution (BD Biosciences, 51-2090KZ) for 30 min at 4 °C, except for simultaneous analysis of aCasp3$^+$ and FRET$^{neg}$ cells where fixation for 20 min at room temperature was found to better preserve INDIA fluorescence. Cells were washed twice with Perm/Wash buffer (BD Biosciences, 51-2091KZ) and then incubated with antibodies against intracellular antigens (see Supplementary Table 1) diluted in Perm/Wash buffer for 45 min at 4 °C. After three washes with Perm/Wash buffer, cells were resuspended in FACS buffer.

Cells were acquired on LSRII and LSRFortessa flow cytometers (BD Biosciences). Rosa26$^{INDIA}$ cells were acquired on LSRFortessa and FACSymphony A5 flow cytometers (BD Biosciences) with the following specifications: mNeonGreen (488 nm excitation, 530/30 BP, 505 LP), FRET (488 nm excitation, 610/20 BP, 600 LP), mRuby2 (561 nm excitation, LSRFortessa: 582/15 BP; FACSymphony: 585/15 BP, 570 LP). Rosa26$^{INDIA}$ bulk cell sorting was performed on FACSAria and Fusion instruments (BD Biosciences) with the same specifications indicated above for LSRFortessa. Flow cytometry data were analyzed using FlowJo software versions 10.7, 10.8 or 10.9 (BD Biosciences). Doublets were excluded using FSC-W. For Rosa26$^{INDIA}$ analysis and cell sorting, FRET$^{loss}$ was derived as new parameter by dividing mNeonGreen and FRET signals[17].

Gating for Rosa26$^{INDIA}$ experiments was performed as follows:

In Figs. 1 and 3, bone marrow B cells were gated CD45R/B220$^+$mRuby2$^+$DAPI$^{neg}$ and CD43$^+$IgD$^{neg}$IgM$^{neg}$ (Fr. B/C), CD43$^{neg}$IgD$^{neg}$IgM$^{neg}$ (Fr. D), CD43$^{neg}$IgD$^{neg/lo}$IgM$^+$ (Fr. E) or CD43$^{neg}$IgD$^{hi}$IgM$^+$ (Fr. F). Blood and spleen B cells were gated CD45R/B220$^+$mRuby2$^+$DAPI$^{neg}$Lineage(CD4, CD8α, F4/80, NK1.1, Ly-6G, Ter-119)$^{neg}$CD95$^{neg}$ and IgD$^{lo}$IgM$^{hi}$CD21$^{neg/lo}$ (transitional, T), IgD$^{lo}$IgM$^{hi}$CD21$^{neg}$CD23$^{neg}$ (transitional 1, T1), IgD$^{lo}$IgM$^{hi}$CD21$^{lo}$CD23$^+$ (transitional 2, T2), IgD$^{lo}$IgM$^{hi}$CD21$^{hi}$ (MZ), IgD$^+$IgM$^+$CD23$^+$CD93/AA4.1$^{neg}$ (follicular, FO), or IgD$^+$IgM$^+$CD23$^+$CD93/AA4.1$^+$IgM$^{low}$ (anergic, T3). The full gating strategy is shown in Supplementary Fig. 1c–e. We recommend avoiding CD93/AA4.1 if possible because this marker is relatively weakly expressed on transitional B cells and tends to be further downregulated during apoptosis resulting in incomplete capturing of all FRET$^{neg}$ cells.

Gating in Fig. 2 was performed as in Fig. 1 except intravascular anti-CD45R/B220-PE-Cy7 was used to label bone marrow sinusoids, CD19 was used to stain B cells instead of CD45R/B220, dead cells were excluded using Zombie NIR, and FO B cells were gated IgD$^{hi}$IgM$^+$.

Gating in Supplementary Fig. 1j, k and Supplementary Fig. 2 is as in Fig. 1 but without CD93/AA4.1 and distinguishing T3 cells.

Gating in Supplementary Fig. 4 was CD45R/B220$^+$mRuby2$^+$DAPI$^{neg}$Lineage(CD4, CD8α, F4/80, NK1.1, Ly-6G, Ter-119)$^{neg}$CD95$^{neg}$ and CD93/AA4.1$^+$CD23$^{neg}$ (transitional 1 B cells, T1) or CD93/AA4.1$^{neg}$CD23$^+$ (mature follicular B cells, FO). IgM$^{lo}$ and IgM$^{hi}$ subsets were distinguished.

For Rosa26$^{INDIA}$ bulk sorting in Supplementary Fig. 1b, the following gating was done on bone marrow and spleen cells: CD45R/B220$^+$Lineage(CD4, CD8α, NK1.1, F4/80, Ly-6G, Ter-119)$^{neg}$DAPI$^{neg}$mRuby2$^+$ and FRET$^+$ or FRET$^{neg}$. In case of splenic B cells, GL7$^{hi}$ GC B cells were additionally excluded.

Gating for other experiments was performed as follows:

In Fig. 5, spleen B cells were gated PI$^{neg}$CD138$^{lo}$TACI$^{lo}$CD19$^+$Fas$^{neg}$ (Fig. 5a–d) or Zombie-NIR$^{neg}$Lin(CD4,CD8α,NK1.1,Ly-6G,Ter-119,F4/80)$^{neg}$CD19$^+$Fas$^{neg}$ (Fig. 5e–j) and CD93/AA4.1$^+$CD21$^{neg}$CD23$^{neg}$ (transitional 1 B cells, T1), AA4.1$^+$CD21$^{lo}$CD23$^+$ (transitional 2/3 B cells, T2/3) or CD93/AA4.1$^{neg}$CD21$^+$CD23$^+$ (mature follicular B cells, FO). For FO B cells in Fig. 5a–d, GL7$^+$ cells were additionally excluded. See Supplementary Fig. 5 for gating.

## Single B cell sorting, antibody cloning and recombinant expression

Euthanized Rosa26$^{INDIA}$ mice, buffers, and consumables contacting cells were pre-chilled and maintained on ice to minimize de novo apoptosis during cell isolation, staining and cell sorting. Centrifugation and cell sorting were done at 4 °C. Typically, two mice were pooled per sort. Staining was performed as described in the Flow cytometry section. Bone marrow cells were gated CD45R/B220$^+$Lineage (CD4, CD8α, NK1.1, F4/80)$^{neg}$Zombie-NIR$^{neg}$mRuby2$^+$CD43$^{neg}$ and IgM$^{neg}$IgD$^{neg}$ (Fr. D) or IgM$^+$IgD$^{neg/lo}$ (Fr. E). For one Fr. D sort, bone marrow cells were sorted separately from two individual mice, dead cells were excluded with DAPI and Lineage additionally included Ly-6G/Ly-6C and Ter-119. B lymphocytes were enriched from erythrocyte-lysed spleen cells using CD43 MicroBeads (Miltenyi Biotec,130-049-801) and LS columns (Miltenyi Biotec, 130-042-401) according to the manufacturer's instructions prior to staining. Spleen cells were gated CD19$^+$Lineage(CD4, CD8α, NK1.1, F4/80)$^{neg}$Zombie-NIR$^{neg}$mRuby2$^+$Fas$^{neg}$GL7$^{lo}$ and IgD$^{lo}$IgM$^{hi}$CD21$^{lo}$ (transitional) or IgD$^{hi}$IgM$^+$ (mature follicular B cells, FO). For each B cell population, FRET$^+$ and FRET$^{neg}$ single B cells were sorted into 96-well PCR plates containing 4 µl lysis buffer (for composition see[52]) and were immediately frozen on dry ice. Plates were directly processed or stored at -80 °C. cDNA preparation, Ig gene amplification, sequencing and cloning into human expression vectors, recombinant expression and antibody purification were essentially done by following published protocols[17,28,52] except that only IgM-specific reverse primers were used for IgH amplification, and optimized primers were used for Igκ and Igλ sequencing after the 2$^{nd}$ PCR (for a list of all primers used see Supplementary Data 2). Only cells yielding paired Ig bands of the correct size on gel electrophoresis and subsequent high-quality DNA sequencing were considered further. If N bases were called in a high-quality sequence, traces were inspected manually and corrected if the base pair could be clearly assigned. Ig sequences were analyzed with IMGT/V-QUEST and IgBlast and were directly synthesized for cloning (IDT). Fr. D cells carrying a functional BCR were called early immature B cells[29]. 293-F cells (Thermo Fisher, R79007) were maintained in 293 Freestyle medium (Thermo Fisher, 12338026) and were transiently co-transfected with IgH and IgL expression vectors for 7 days prior to monoclonal antibody purification from the culture supernatant using Protein G Sepharose 4 Fast Flow beads (Cytiva,17-0618-05) and IgG elution buffer (Thermo Fisher, 21009). After neutralization and buffer exchange to PBS, the IgG concentration was determined using Nano-Drop One (Thermo Fisher, ND-ONE-W). All monoclonal antibodies were assessed for integrity by SDS-PAGE analysis.

## FLOWMIST assay for autoreactivity

Spleen cells were prepared for intracellular staining as described under Flow cytometry with the following modifications: cells were stained with the Live/Dead Fixable Aqua Dead Cell Stain Kit (Invitrogen, L34966) for 30 min at 4 °C (1:500 dilution in PBS) prior to Fc blocking. Surface staining with anti-CD3ε-PE (hamster IgG, not recognized by the secondary antibody used in later steps) was additionally performed. Cells were fixed using the Foxp3/Transcription Factor Staining Buffer concentrate and diluent (Thermo Fisher Scientific, 00-5523-00) for

30 min at 4 °C. Fixed cells were washed twice with 1xPerm/wash buffer (BD, 554723) or with 1xPermeabilization Buffer (Thermo Fisher Scientific, 00-8333-56). Monoclonal antibodies were diluted to 10 µg/ml in 1xPerm/wash buffer or 1xPermeabilization Buffer and cells were incubated for 1 h at 4 °C followed by three wash steps. Bound autoreactive antibodies were detected with 1 µg/ml AlexaFluor647-conjugated F(ab)2 goat anti-human IgG Fc (Jackson ImmunoResearch, 109-606-098). Secondary staining was performed in the same manner as primary staining. After another three washes, cells were resuspended in FACS buffer and acquired on a BD LSRFortessa X-20 Cell Analyzer (BD Biosciences). Mean AlexaFluor647 fluorescence intensity was calculated for $Aqua^{neg}CD3\varepsilon^{+}$ gated T cells. The MFI ratio was calculated relative to mGO53 negative control monoclonal antibody. Monoclonal antibodies were deemed autoreactive if the MFI ratio was greater than 3.

Autoreactive monoclonal antibodies were further analyzed for the fluorescence pattern as follows. Intracellular staining was performed as described above but without Aqua staining. After secondary antibody incubation, cells were stained with 5 µg/ml DAPI for 5 min at room temperature prior to washing three times. For imaging flow cytometry analysis, cells were resuspended in FACS buffer and acquired on Amnis ImageStream^x MKII using INSPIRE (v200.1.681.0) software at 60x magnification. Images were analyzed using IDEAS v 6.3 software and standard ImageStream analysis wizards and tools/algorithms. For confocal microscopy, cells were resuspended in PBS and transferred to a 35 mm dish with glass coverslip (Ibidi, 81156). Cells were incubated at room temperature for 10 min, then PBS was aspirated and 400 µl fresh PBS were added to cover cells. Cells were imaged with 60x magnification on a Nikon Ti2/Yokogawa CSU-W1 spinning disk confocal microscope with a Hamamatsu Orca Flash 4.0 camera. Confocal images were analyzed using ImageJ.

### Dot blot assay for autoreactivity

The dot blot assay was adapted from[53]. The following lysis buffer was prepared freshly and kept on ice: 1xTBS (Thermo Fisher, 28358) supplemented with 1% NP-40 (Sigma, I8896), Complete Protease Inhibitor Cocktail (Roche, 11873580001) and 1xHalt Phosphatase Inhibitor Cocktail (Thermo Fisher, 78428). Spleen, kidney, and thyroid of C57Bl/6 J mice were cut into small pieces. About 100 mg tissue was frozen in liquid nitrogen, then placed on ice and submerged in 4 ml lysis buffer. Bone marrow was directly flushed out from two femurs and two tibiae with 4 ml lysis buffer. Tissue was homogenized on ice for 1 min on medium setting using a Fisherbrand 150 Homogenizer (Fisher Scientific, 15-340-167) and Plastic disposable generator probes (Fisher Scientific, 15-340-176). Lysates were agitated for 2 h on ice, centrifuged for 20 min at 18,845xg and the supernatant was harvested and stored at -80 °C. Protein concentrations were determined with the Pierce BCA Protein Assay Kit (Thermo Fisher, 23225) according to the manufacturer's instructions. Dot blots were performed using Nitrocellulose membranes (Thermo Fisher, 88018) and the Bio-Dot Microfiltration Apparatus (Bio Rad, 1706545) according to the manufacturer's instructions. In brief, 1 µg lysate was loaded per well in 1xTBS, and wells were then blocked for 2 h at room temperature using 2% (w/v) BSA in 1xTBS. After washing with 1xTBS 0.05% (v/v) Tween20, monoclonal antibodies were incubated at 10 µg/ml for 2 h at room temperature. After washing, secondary peroxidase-conjugated goat anti-human IgG Fc (Jackson ImmunoResearch, 109-035-098) was incubated at room temperature for 1 h at 1:5000 dilution in blocking buffer containing 0.05% (v/v) Tween20. After washing, the membrane was removed from the apparatus, washed in 1x TBS and then incubated for 5 min with SuperSignal West Pico PLUS Chemiluminescent Substrate (Thermo Fisher, 34577). The chemiluminescent signal was scanned with UVP ChemStudio (Analytik Jena). Images were analyzed with ImageJ.

### Statistics and reproducibility

Statistical significance was determined using GraphPad Prism software (version 10). Data were first evaluated for normal distribution by Anderson-Darling test, D'Agostino & Pearson test, Shapiro-Wilk test and Kolmogorov-Smirnov test. If any test reported N is too small for evaluation or if data were normally distributed, then unpaired two-tailed t-test was used. If data did not pass normality tests, unpaired two-tailed Mann-Whitney test was used. Pie charts were compared using two-tailed Fisher's exact test. No adjustments were made for multiple comparisons. Test results are indicated in the Figures and Figure legends. Experiments were performed two to three times independently or using two independent methods with similar results. No statistical method was used to predetermine sample size. No data were excluded from the analyses unless indicated in the methods or figure legends. The experiments were not randomized. The Investigators were not blinded to allocation during experiments and outcome assessment.

### Reporting summary

Further information on research design is available in the Nature Portfolio Reporting Summary linked to this article.

### Data availability

The BCR sequencing data have been deposited in the NLM/NCBI/SRA under accession code PRJNA1054624. Properties of antibodies reported in this study are summarized in Supplementary Data 1. Source data are provided with this paper.

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

## Acknowledgements

We thank all members of the Experimental Immunology Branch, particularly Alfred Singer, Hyun Park, Richard Hodes, Vanja Lazarevic, Paul Roche, Susan Sharrow for critical discussions and advice; Jeffrey Chiang for technical support; Assiatu Crossman, Larry Granger, Tony Adams, William Hajjar, Tengfei Zhang for flow cytometry support; Jan Wisniewski for microscopy support. We thank Jagan Muppidi for discussions, and all staff at the NCI Frederick and NCI Bethesda animal facilities for their

critical help, particularly Jennifer Wise. This work was supported by the Intramural Research Program of the National Cancer Institute, Center for Cancer Research, National Institutes of Health (ZIA BC011975, C.T.M.). This research was supported in part by the Intramural Research Program of the NIH, National Institute of Allergy and Infectious Diseases (I.D.), and the National Natural Science Foundation of China (32070947, Q.W.). C.T.M. is a Stadtman Investigator. M.J.S., S.S., A.M.N. and D.P. are CRTA fellows. Additionally, the authors thank Yolanda L. Jones, NIH Library Editing Services, for editing assistance.

## Author contributions

M.J.S. performed research, collected data, analyzed, and interpreted data, performed statistical analysis, and wrote the manuscript. A.M.N. performed research, collected data, analyzed, and interpreted data and performed statistical analysis. C.M.N. and D.P. performed research. S.S. performed research and collected data. T.M., D.S. and I.D. performed research and contributed vital new analytical tools. Q.W. contributed vital analytical tools and analyzed data. C.T.M. designed research, performed research, collected data, analyzed, and interpreted data, performed statistical analysis, and wrote the manuscript.

## Funding

## Competing interests

The authors declare no conflicts of interest.
