## [Peer Review File · Nature Communications]

Peripheral apoptosis and limited clonal deletion during physiologic murine B lymphocyte developmentREVIEWER COMMENTS

Reviewer #1 (Remarks to the Author):

The manuscript by Simpson et al. reports that clonal deletion of self-reactive B cells by apoptosis is not a major mechanism of central B cell tolerance in the mouse bone marrow, as is generally believed. Instead, apoptosis was detected primarily in the transitional B cells in the periphery, however, it was not associated with autoreactivity. Thus, this work will significantly contribute to the field of B cell tolerance by defining a lack of apoptosis contribution to this process. Because apoptosis is difficult to prove experimentally due to the quick clearance of dead cells, the contribution of apoptosis to immune tolerance remains enigmatic. While the question of whether and to what extent apoptosis contributes to B cell tolerance is not original, the authors' approach is cutting-edge and appears to provide novel insights into this longstanding question and show that clonal deletion appears to make a minor contribution to apoptosis detectable during physiologic B cell development. However, this should be interpreted with caution because it is uncertain whether all or even most apoptotic B cells can be detected by the FRET^{neg} indicator due to the short existence of cells undergoing apoptosis. Even with the consideration of the limits of FRET approach this study will have high impact on the field of B cell tolerance.

The findings in this manuscript using a FRET^{neg} fluorescent indicator of apoptosis by caspase-3 cleavage (ROSA26INDIA mice), reveal that apoptosis is generally low in the bone marrow, with the highest in the Pre-B (fraction D), reduced in the immature (IgM⁺) B, and minimal in the recirculating mature follicular B cells. Unexpectedly, transitional B cells in the blood and spleen had the highest level of apoptosis, which was reduced with maturation. Thus, this work challenges the previous findings by *in vivo* BrdU labeling (Allman and Cancro 1993), which concluded that apoptosis significantly contributes to immature B cell central tolerance in the bone marrow.

The experiments are generally of high quality and use appropriate and cutting-edge technology (ROSA26INDIA mouse model) and the interpretations and conclusions are generally supported by the data. However, the data interpretation required some necessary assumptions inherent to the apoptosis field, namely the detection of dying cells, but these were addressed by the authors. Nonetheless, these caveats limit conclusions.

Apoptosis was analyzed during physiologic B cell development in the bone marrow and during continued selection and maturation in the periphery (blood and spleen), which is a strength of the paper. They quantified apoptosis using an innovative experimental mouse model, Rosa26INDIA mice, which they have previously described to quantify apoptosis among germinal center B cells. In these mice, the fluorescence indicated by caspase-3 cleavage and thus caspase activation could be detected by flow cytometry. The FRET^{neg} cells were assigned as apoptotic. The authors also explored the apoptosis of immature B cells upon release from bone marrow sinusoids into the blood circulation, demonstrating a dramatic increase in apoptosis upon dissociation from the BM sinusoids potentially due to lack of survival signals. The authors did a good job in defining all the B cell populations by FCM. They also cloned, expressed and tested BCRs from the isolated apoptotic B cells for autoreactivity, a significant strength of this study.

A concern is the reliability of the DNA sequences of the genomic DNA isolated from the apoptotic cells. Isn't DNA fragmentation a hallmark of apoptosis? Given this, can the V(D)J sequencing of the BCRs be reliable? The other concern is whether the apoptotic cells analyzed are sufficiently representative of actual apoptosis and whether the data can be used to make sound conclusions.

Specific comments:

1. A table with all the percentages for various B cell populations will increase comprehension of the data.
2. The authors should discuss a previous study by the Marrack group, which showed the importance of apoptosis in BM B cell tolerance using Bim-deficient BM B cells.
3. The authors should analyze T2 and anergic B cells for apoptosis and autoreactivity. Their apoptosis and autoreactivity could serve as a reference population. There is more apoptosis in anergic B cells because they have a shorter life span and increased dependence on BAFF survival signals. This is relevant to authors' conclusion that lack of survival signaling rather than clonal deletion appears to underlie most transitional B cell apoptosis in the periphery.

4. Explain if the anergic B cells were excluded from the mature B cells and if not what is their contribution to apoptosis and autoreactivity.
5. Is all/most B cell apoptosis dependent on Caspase3?
6. Are the B cell clones with autoreactive BCR engaged with the autoAgs and at a similar level in the BM and periphery?

Although, the manuscript is well written, at places additional information or explanation would benefit the Nature Communications readership. The manuscript will improve by providing

7. More explanation of the mouse model
8. Further information about the previous work using BrdU
9. If possible, discuss whether the data in this work is in conflict with the previous work or can be explained by this work.

Reviewer #2 (Remarks to the Author):

How self-reactive and polyreactive B cells generated by VDJ recombination during B-cell development are eliminated, or silenced, is an important question and is incompletely understood. Whether (and at which developmental stage) immunoreactive B-cells undergo apoptosis, receptor editing or anergy remains unclear.

Here the authors have carefully tracked the appearance of apoptotic B-cells from the bone marrow to the periphery throughout development and argue, contrary to the dominant view, that receptor editing rather than clonal deletion eliminates the majority of self-reactive B cells in the bone marrow, with much higher rates of apoptosis observed in peripheral B-cells at the transitional stage. They further suggest that lack of survival signaling, resulting in loss of the critical survival protein Bcl-2, rather than clonal deletion underlies most transitional B-cell apoptosis in the periphery.

In general the work is carefully performed and discussed and the analyses are thorough and well explained.

I have the following suggestions for improvement.

1. The authors use ROSA26INDIA mice expressing a reporter of caspase-3 activation, which most of the analyses are heavily reliant upon. It's important that the authors demonstrate that this reporter is a sufficiently sensitive indicator of apoptosis by conducting the analyses presented in Figure 1 (at least on peripheral blood and splenic B-cells) with the more well established FLICA-VAD-fmk poly-caspase labelling probe to confirm that the casp-3 FRET probe they are using is sufficiently sensitive to detect all apoptotic B-cells.

2. There is an underlying assumption (however well founded) that B-cell deletion necessarily occurs in a caspase-3-dependent manner (i.e. via apoptosis) but this might not be the case. I feel that this is especially important to clarify given that the authors found (Figure 4) that there was surprisingly little difference in poly-reactivity or self-reactivity between BCRs cloned from FRET+ versus FRET- B-cells at most stages of development. For further reassurance that all developmental B-cell death is effectively detected using the FRET probe or the VAD-FLICA probe, the authors should also compare the percentages of cell death they observe in peripheral blood and splenic B-cells using the FRET probe, FLICA-VAD versus data obtained with fluorescently-tagged Annexin V. In addition to binding to apoptotic cells, annexin V will also detect PS externalization due to other modes of cell death (necroptosis, pyroptosis, ferroptosis, necrosis). This would eliminate the concern that a proportion of the B-cell death occurring in the BM or other compartment could be via an alternative mode of programmed cell death that is caspase-independent (or caspase-3-independent).

3. Although the authors have injected Dex to induce B-cell death polyclonally and have shown that there is a dramatic rise in FRET negative cells as a consequence (supp Fig. 1), it would be

instructive to see the results of this analysis comparing the % of FRET negative B-cells versus VAD-FLICA+ and Annexin V+ cells, once again to exclude the possibility that negative selection could occur in a caspase-3-independent manner. It would also be useful to trigger B-cell death somewhat more selectively through use of a B-cell superantigen or anti-BCR antibodies.

Point-by-point response

We thank the Reviewers for their overall enthusiastic evaluation, thoughtful comments, and helpful suggestions. The manuscript has been revised in response to the Reviewers' comments which has improved the overall quality of the manuscript. We believe that all concerns have been addressed. A detailed point-by-point response is provided below.

Reviewer #1:

A concern is the reliability of the DNA sequences of the genomic DNA isolated from the apoptotic cells. Isn't DNA fragmentation a hallmark of apoptosis? Given this, can the V(D)J sequencing of the BCRs be reliable?

Our V(D)J sequencing relies on mRNA which is converted to cDNA and then PCR amplified for each Ig gene, not genomic DNA. Thus, genomic DNA fragmentation would not affect our V(D)J sequencing. However, the reviewer is right because even mRNA appears to become compromised during apoptosis because the cloning efficiency of FRET^{neg} B cells is lower than that of FRET⁺ B cells. This requires processing more FRET^{neg} B cells than FRET⁺ B cells, further illustrating the effort and time that went into these experiments. Importantly, our data only include B cells that harbored sufficient high-quality mRNA that allowed for successful PCR amplification of the paired Ig genes. Additionally, among those cells with paired Ig bands of the correct size on gel electrophoresis, only those yielding high-quality DNA sequences were included in our analyses. We have added these important details to the revised methods section. In summary, we believe that the reported V(D)J sequences are reliable.

The other concern is whether the apoptotic cells analyzed are sufficiently representative of actual apoptosis and whether the data can be used to make sound conclusions.

Thank you for this important point. We would first like to refer to Supplementary Fig. 1b where we show that FRET⁺ FACS-sorted B cells contain little aCasp3⁺ cells whereas >94% of FACS-sorted FRET^{neg} B cells are aCasp3⁺. This demonstrates that the FRET probe is sensitive enough to detect the vast majority of aCasp3⁺ apoptotic B cells.

We also addressed this point directly by comparing intracellular aCasp3 staining and FRET^{neg} cells in the same samples for all B cell subsets. In addition to the steady state, we also induced apoptosis *in vitro* by brief culture at 37°C. The data are shown in the new Supplementary Fig. 1i-k. Intracellular aCasp3 staining reproduced our findings of low apoptosis in bone marrow B cell subsets and increased levels of apoptosis in splenic T1 and T2 cells. Moreover, culture at 37°C induced apoptosis in all B cell subsets compared with control cells, and simultaneous measurements of aCasp3⁺ cells and FRET^{neg} cells in the same samples were comparable. The results have been updated to incorporate the new data.

I would also like to highlight that cloning and producing 248 monoclonal antibodies (at least 30 per compartment) was a major effort and only made technically possible by using Rosa26^{INDIA} mice. While we would prefer to assay as many B cells as possible, producing at least 30 antibodies from each compartment is a feasible starting point and usually

sufficient to test if major differences exist between compartments. Moreover, the FRET^{neg} cells analyzed by single-cell cloning were randomly selected.

In summary, we believe that the FRET^{neg} cells analyzed are sufficiently representative of actual aCasp3-mediated apoptosis and allow sound conclusions.

Specific comments:

1. A table with all the percentages for various B cell populations will increase comprehension of the data.

We have created this table for the measurements in Fig. 1 and now refer to it as Table 1 in the results section. We also created a table for the comparisons in Supplementary Fig. 2 and now refer to it as Table 2 in the results section.

2. The authors should discuss a previous study by the Marrack group, which showed the importance of apoptosis in BM B cell tolerance using Bim-deficient BM B cells.

Thank you, we have included this study into our discussion.

3. The authors should analyze T2 and anergic B cells for apoptosis and autoreactivity. Their apoptosis and autoreactivity could serve as a reference population. There is more apoptosis in anergic B cells because they have a shorter life span and increased dependence on BAFF survival signals. This is relevant to authors' conclusion that lack of survival signaling rather than clonal deletion appears to underly most transitional B cell apoptosis in the periphery.

To address this point, we revised the gating strategy in Supplementary Fig. 1d and e and now separate anergic (T3) B cells from mature follicular (FO) B cells. Fig. 1c-f and the results have been revised accordingly and we now compare apoptosis of T1, T2, anergic (T3) and mature follicular (FO) B cells. Blood and spleen T3 cells contained significantly more FRET^{neg} cells (average 0.265% to 0.284%) than FO B cells (average 0.112% to 0.132%; Fig. 1f and Table 1), consistent with a shorter life span of anergic B cells. Notably, apoptosis of T3 cells was still markedly lower than that of T1 cells. We now also discriminated T3 cells from FO B cells in the revised Fig. 3 and updated the numbers accordingly.

While we agree that determining the autoreactivity of FRET⁺ and FRET^{neg} T2 and T3 cells would nicely complete our analyses, I would like to highlight the time-consuming and difficult nature of these experiments that would involve new single-cell sorts, PCRs, cloning, production, and purification of at least 120 new antibodies, and performance of the three autoreactivity assays described in the paper. Please also note our response to your first point: because mRNA gets degraded during apoptosis, the cloning of FRET^{neg} B cells requires processing more samples than for FRET⁺ B cells. We estimate this experiment alone could take around one year to complete. Given the dramatic peak of apoptosis in T1 cells, we think that, in the scope of this study, it was reasonable to focus our workforce and resources on the T1 population. Of note, Merrell et al. (PMID: 17174121) compared autoreactivity of T1, T2 and T3 cells after stimulating polyclonal antibody secretion with LPS. Furthermore, Nojima et al. (PMID: 32414809) have

compared autoreactivity among early immature B cells, immature B cells, and T3 cells using single cell cultures. We have now discussed these findings in the revised version.

4. Explain if the anergic B cells were excluded from the mature B cells and it not what is their contribution to apoptosis and autoreactivity.

In the old manuscript version, mature FO B cells in Fig. 1 and Fig. 3 originally included anergic B cells. To address this point in the revised version, we changed the gating strategy in Supplementary Fig. 1d and e and now separate anergic (T3) B cells from mature follicular (FO) B cells. Fig. 1c-f has been revised accordingly and we now compare apoptosis of T1, T2, anergic (T3) and mature follicular (FO) B cells. Blood and spleen T3 cells contained significantly more FRET^{neg} cells (average 0.265% to 0.284%) than FO B cells (average 0.112% to 0.132%; Fig. 1f and Table 1), consistent with a shorter life span of anergic B cells. Notably, apoptosis of T3 cells was still markedly lower than that of T1 cells. Fig. 3 has also been revised and now discriminates T3 and FO B cells. Given that T3 cells make up about 7% of FO splenic B cells (Merrell et al.; PMID: 17174121), T3 cells would make a minimal contribution to the apoptosis of FO B cells if the two subsets were not distinguished.

As for single cell sorting, we did not differentiate T3 from FO B cells. It is therefore possible that T3 cells contributed to the autoreactivity detected among FO B cells in Fig. 4. To examine this, we considered that 10% (Merrell et al.; PMID: 17174121) to 30% (Nojima et al.; PMID: 32414809) of T3 cells are autoreactive. Given that T3 cells make up about 7% of mature splenic B cells (Merrell et al.; PMID: 17174121), we estimate that T3 cells could have contributed 0.7% to 2.1% of autoreactivity to FO B cells, markedly below our measured number of 10%. In summary, based on published data and our measurements of FRET^{neg} and autoreactive cells, we think that T3 cells make a small contribution to apoptosis and autoreactivity of FO B cells if the two subsets are not distinguished. This has now been pointed out in the revised discussion.

5. Is all/most B cell apoptosis dependent on Caspase3?

Strasser, Takeguchi and Wright et al. have independently shown that defective mitochondrial apoptosis results in increased numbers of several B cell subsets (PMID: 1924327; PMID: 16055554; PMID: 34512628). Additionally, defective death receptor-mediated apoptosis, for example downstream of Fas, results in lymphoproliferation (PMID: 18835195). No such effects on B cells have been reported in mice with defective aCasp3-independent cell death (e.g. pyroptosis and necroptosis). It is therefore quite well established that most steady state B cell apoptosis requires either the mitochondrial apoptosis pathway or the death receptor apoptosis pathway both of which require aCasp3.

Nevertheless, we addressed this point by using a FLICA-VAD probe that detects all active caspases, and Annexin-V staining which detects phosphatidylserine exposed during late stages of apoptosis and other types of cell death. These results are shown in the new Supplementary Fig. 2a-h. We observed more FLICA-VAD⁺ and Annexin-V⁺ cells than FRET^{neg} and aCasp3⁺ cells across most B cell subsets and have revised the results and

discussion accordingly. It will be interesting to investigate aCasp3-independent cell death in future studies to rule out or establish a contribution to B cell development.

To conclude, B cell apoptosis requires aCasp3 and the results and conclusions of the present study are focused on detecting aCasp3-dependent apoptosis.

6. Are the B cell clones with autoreactive BCR engaged with the autoAgs and at a similar level in the BM and periphery?

IgM BCR engagement by autoAgs results in IgM downregulation. We retrospectively analyzed IgM expression on our cloned FRET+ B cells at the time of Index single-cell sorting as an indicator of autoAg engagement and compared autoreactive/polyreactive to non-reactive B cells (see Figure below). Early immature B cells where most self-reactivity is found could not be analyzed since they are surface IgM^{neg} by definition. In the bone marrow, 2 of 3 polyreactive/autoreactive immature B cells had markedly lower IgM expression than the average population, whereas 4 of 5 autoreactive/polyreactive B cells in the periphery had IgM expression closer to the average. This would argue that self-reactive immature B cells in the bone marrow strongly engage autoAgs consistent with a significant removal of polyreactive/autoreactive B cells at this site. Polyreactive/autoreactive B cells that further mature and are found in the periphery appear to engage autoAgs to a lesser extent in the periphery. Because Fig. 4 and Supplementary Fig. 3 related to polyreactivity/autoreactivity measurements are already very packed, and because the numbers of polyreactive/autoreactive clones that we can analyze here are quite low, we prefer not to include these data into the revised manuscript. However, we revised the description of our data in the manuscript to include statements about engagement with autoAgs in the bone marrow and periphery.

Analysis of IgM expression among cloned B cells. Cloned B cells were categorized into polyreactive (red), autoreactive (green) or nonreactive (gray) according to our monoclonal antibody testing results. The IgM mean fluorescence intensity is plotted against SSC-A for all cloned FRET+ bone marrow immature B cells, splenic transitional B cells or splenic follicular B cells. The upper dotted line represents the average IgM expression for each population. The lower dotted line is the average IgM expression of bone marrow Fraction D cells which do not express surface IgM.

Although, the manuscript is well written, at places additional information or explanation would benefit the Nature Communications readership. The manuscript will improve by providing

7. More explanation of the mouse model

We added a paragraph to the Introduction.

8. Further information about the previous work using BrdU

We added more information about the previous BrdU experiments.

9. If possible, discuss whether the data in this work is in conflict with the previous work or can be explained by this work.

Our data agree with previous work by several laboratories showing that the bone marrow microenvironment suppresses immature B cell apoptosis and that immature B cells do not frequently undergo clonal deletion in the bone marrow. These results differ from the interpretations of previous BrdU experiments regarding B cell apoptosis in the bone marrow. We have provided more background to these BrdU experiments and discussed potential reasons for the discrepancy. These include interpretations of the previous BrdU experiments, limitations and assumptions of our experimental approach, and possible aCasp3-independent cell death.

Reviewer #2:

I have the following suggestions for improvement.

1. The authors use ROSA26INDIA mice expressing a reporter of caspase-3 activation, which most of the analyses are heavily reliant upon. It's important that the authors demonstrate that this reporter is a sufficiently sensitive indicator of apoptosis by conducting the analyses presented in Figure 1 (at least on peripheral blood and splenic B-cells) with the more well established FLICA-VAD-fmk poly-caspase labelling probe to confirm that the casp-3 FRET probe they are using is sufficiently sensitive to detect all apoptotic B-cells.

Thank you for this important point. We would first like to refer to Supplementary Fig. 1b where we show that FRET⁺ FACS-sorted B cells contain little aCasp3⁺ cells whereas >94% of FACS-sorted FRET^{neg} B cells are aCasp3⁺. This demonstrates that the FRET probe is sensitive enough to detect the vast majority of aCasp3⁺ apoptotic B cells.

We also addressed this point directly by comparing intracellular aCasp3 staining and FRET^{neg} cells in the same samples for all B cell subsets. In addition to the steady state, we also induced apoptosis *in vitro* by brief culture at 37°C. The data are shown in the new Supplementary Fig. 1i-k. Intracellular aCasp3 staining reproduced our findings of low apoptosis in bone marrow B cell subsets and increased levels of apoptosis in splenic T1 and T2 cells. Moreover, culture at 37°C induced apoptosis in all B cell subsets compared with control cells, and simultaneous measurements of aCasp3⁺ cells and FRET^{neg} cells in the same samples were comparable. The results have been updated to incorporate the new data. Notably, intracellular aCasp3 staining was not sensitive enough to reveal differences in B cell subsets exhibiting low levels of apoptosis including bone marrow B cell subsets, splenic FO, and splenic MZ B cells, which was easily possible by quantitating FRET^{neg} cells in the same samples even after fixation.

We have also performed staining with the FLICA-VAD polycaspase probe on bone marrow and spleen B cell subsets, mostly with the goal to address aCasp3-independent cell death. Since we performed three different stains simultaneously under two different conditions per mouse, we decided not to include blood because the numbers of B cells in each sample would be too limited. The results are shown in the new Supplementary Fig. 2a and 2b and in Table 2. Cells from untreated Rosa26^{INDIA} mice stained at 4°C (control) had approximately 3 to 26 times more FLICA-VAD⁺ than FRET^{neg} and aCasp3⁺ B cells except for splenic T1 cells that were comparable. When performing FLICA-VAD staining at 37°C which is the manufacturer's recommended protocol, there were also 3 to 21 times more FLICA-VAD⁺ than FRET^{neg} and aCasp3⁺ B cells under the same conditions except for splenic T1 cells that were comparable. These results could point towards caspase activity other than aCasp3 in most B cell subsets but without any marked regulation during B cell development. The results and discussion have been updated accordingly. It will be interesting to investigate aCasp3-independent cell death in future studies to rule out or establish a contribution to B cell development.

Collectively, we have demonstrated that Rosa26^{INDIA} mice sensitively detect all aCasp3⁺ apoptotic B cells and the results and conclusions of the present study are focused on detecting aCasp3-dependent apoptosis.

2. There is an underlying assumption (however well founded) that B-cell deletion necessarily occurs in a caspase-3-dependent manner (i.e. via apoptosis) but this might not be the case. I feel that this is especially important to clarify given that the authors found (Figure 4) that there was surprisingly little difference in poly-reactivity or self-reactivity between BCRs cloned from FRET⁺ versus FRET⁻ B-cells at most stages of development. For further reassurance that all developmental B-cell death is effectively detected using the FRET probe or the VAD-FLICA probe, the authors should also compare the percentages of cell death they observe in peripheral blood and splenic B-cells using the FRET probe, FLICA-VAD versus data obtained with fluorescently-tagged Annexin V. In addition to binding to apoptotic cells, annexin V will also detect PS externalization due to other modes of cell death (necroptosis, pyroptosis, ferroptosis, necrosis). This would eliminate the concern that a proportion of the B-cell death occurring in the BM or other compartment could be via an alternative mode of programmed cell death that is caspase-independent (or caspase-3-independent).

We have performed Annexin-V staining in bone marrow and splenic B cell subsets. Since we performed three different stains simultaneously under two different conditions per mouse, we decided not to include blood because the numbers of B cells in each sample would be too low. The results are shown in the new Supplementary Fig. 2c-h and in Table 2. Interestingly, these analyses demonstrated that FRET^{neg} B cells in Rosa26^{INDIA} mice are early apoptotic cells, whereas most Annexin-V⁺ B cells are in a late stage of apoptosis or other forms of cell death and have distinct FSC/SSC profiles. In untreated Rosa26^{INDIA} mice there were 4 to 70 times more Annexin-V⁺ than FRET^{neg} and aCasp3⁺ cells across all B cell subsets. In the bone marrow, all B cell subsets had increased percentages of Annexin-V⁺ cells compared with mature recirculating B cells with highest Annexin-V staining in Fraction B/C. Annexin-V staining was also higher in splenic transitional compared with mature B cells.

Strasser, Takeguchi and Wright et al. have independently shown that defective mitochondrial apoptosis results in increased numbers of several B cell subsets (PMID: 1924327; PMID: 16055554; PMID: 34512628). Additionally, defective death receptor-mediated apoptosis, for example downstream of Fas, results in lymphoproliferation (PMID: 18835195). No such effects on B cells have been reported in mice with defective aCasp3-independent cell death (e.g. pyroptosis and necroptosis). It is therefore quite well established that most steady state B cell apoptosis requires either the mitochondrial apoptosis pathway or the death receptor apoptosis pathway both of which require aCasp3.

To conclude, B cell apoptosis requires aCasp3 and the results and conclusions of the present study are focused on detecting aCasp3-dependent apoptosis. It will be interesting to investigate aCasp3-independent cell death in future studies to rule out or establish a contribution to B cell development.

3. Although the authors have injected Dex to induce B-cell death polyclonally and have shown that there is a dramatic rise in FRET negative cells as a consequence (supp Fig. 1), it would be instructive to see the results of this analysis comparing the % of FRET negative B-cells versus VAD-FLICA+ and Annexin V+ cells, once again to exclude the possibility that negative selection could occur in a caspase-3-independent manner. It would also be useful to trigger B-cell death somewhat more selectively through use of a B-cell superantigen or anti-BCR antibodies.

We have performed these experiments with dexamethasone treated Rosa26^{INDIA} mice to compare percentages of FRET^{neg}, FLICA-VAD⁺ and Annexin-V⁺ B cell subsets. The results are shown in the new Supplementary Fig. 2a-h and in Table 2. The percentages of FLICA-VAD⁺ and FRET^{neg} cells after dexamethasone treatment differed less than 2.1-fold across all B cell subsets. In contrast to our analyses of FLICA-VAD⁺ and FRET^{neg} cells, Annexin-V⁺ cells were not substantially increased in any B cell population after dexamethasone treatment. Nevertheless, the percentages of FLICA-VAD⁺, FRET^{neg} and Annexin-V⁺ cells after dexamethasone treatment differed less than 3.2-fold across all B cell subsets except for Fr. B/C cells that contained 5 to 10 times more Annexin-V⁺ cells. Strasser, Takeguchi and Wright et al. have independently shown that defective mitochondrial apoptosis results in increased numbers of several B cell subsets (PMID: 1924327; PMID: 16055554; PMID: 34512628). Additionally, defective death receptor-mediated apoptosis, for example downstream of Fas, results in lymphoproliferation (PMID: 18835195). No such effects on B cells have been reported in mice with defective aCasp3-independent cell death (e.g. pyroptosis and necroptosis). It is therefore quite well established that most steady state B cell apoptosis requires either the mitochondrial apoptosis pathway or the death receptor apoptosis pathway both of which require aCasp3.

To conclude, B cell apoptosis requires aCasp3 and the results and conclusions of the present study are focused on detecting aCasp3-dependent apoptosis. It will be interesting to investigate aCasp3-independent cell death in future studies to rule out or establish a contribution to B cell development.

While also interesting and potentially complementary to the dexamethasone experiments, we chose not to pursue superantigen or anti-BCR approaches to induce apoptosis for the following reasons: (1) Addition of these new *in vivo* treatments to our animal protocol would require lengthy approval applications (e.g. anti-BCR agents would first require testing for mouse pathogens, and the animal protocol would have to be modified which usually involves at least one review and one revision step prior to approval). (2) Superantigens only target specific Ig genes. Identification of the responding B cells within the diverse B cell pool would be difficult. (3) Anti-BCR and superantigen treatments would lead to BCR complex internalization and/or competition with the anti-BCR antibodies used in our flow cytometry panel. This would make it impossible to distinguish bone marrow Fractions D and E which relies on IgM/IgD staining. Use of IgM/IgD/CD21 would also be prohibited, but these markers are essential for identifying B cell subsets in our gating panel. (4) Anti-BCR treatment was shown to induce B cell apoptosis during *in vitro*

settings. Evidence *in vivo* is quite limited. Early experiments used long-term anti-BCR treatment in developing embryos or in young mice to study subsequent suppression of B cell responses to these agents later in life. These experiments were later interpreted to be consistent with anti-BCR induced developmental arrest, receptor editing, and eventually apoptosis if receptor editing is not possible (discussed in Ait-Azzouzene et al.; PMID: 15738053). The effect of acute anti-BCR *in vivo* treatment in adult mice is not well documented. Hobeika recently demonstrated that anti-IgM treatment for 48h *in vivo* did not cause apoptosis of splenic T1 cells but in the contrary promoted their survival (Hobeika et al; PMID: 30127788). Because dexamethasone causes massive apoptosis in all B cell subsets within 4h, we think this is a much better treatment to carry out these experiments. Nevertheless, we have included a second method to induce B cell apoptosis by incubating isolated cell suspensions for 1h at 37°C *in vitro* and measured the percentages of FRET^{neg} and FLICA-VAD⁺ B cells. These data are included in the new Supplementary Fig. 2a and 2b. Notably, *in vitro* culture for 1h at 37°C was as effective at inducing FRET^{neg} cells in all B cell subsets as *in vivo* treatment with dexamethasone. *In vitro* culture was more effective at inducing polycaspase activity in all B cell subsets compared to dexamethasone treatment.

REVIEWERS' COMMENTS

Reviewer #1 (Remarks to the Author):

The authors have addressed my concerns thoughtfully and as best they could. Considering the challenges inherent in molecularly analyzing apoptotic B cells, I commend the authors for undertaking the daunting task of investigating the role of apoptosis in B cell tolerance in mice. While these studies are not definitive in explaining the role of all mechanisms of cell death in B cell tolerance, given the laborious and technically demanding nature of the experimental approach, they undoubtedly enhance our comprehension of aCasp3-dependent B cell death within the realm of B cell tolerance. They also support a more significant role for apoptosis-independent mechanisms of B cell tolerance, particularly receptor editing as proposed by Dr. Pelanda.

Reviewer #2 (Remarks to the Author):

The authors have responded comprehensively to my previous comments. There are interesting differences between the % of FRET negative cells, VAD-FLICA-positive, and annexin V-positive cells, which could indicate different modes of cell death at play, and these have been documented in the supplemental data which I feel is useful. I have no further comments for improvement.

Point-by-point response

Reviewer #1 (Remarks to the Author):

The authors have addressed my concerns thoughtfully and as best they could. Considering the challenges inherent in molecularly analyzing apoptotic B cells, I commend the authors for undertaking the daunting task of investigating the role of apoptosis in B cell tolerance in mice. While these studies are not definitive in explaining the role of all mechanisms of cell death in B cell tolerance, given the laborious and technically demanding nature of the experimental approach, they undoubtedly enhance our comprehension of aCasp3-dependent B cell death within the realm of B cell tolerance. They also support a more significant role for apoptosis-independent mechanisms of B cell tolerance, particularly receptor editing as proposed by Dr. Pelanda.

Reviewer #2 (Remarks to the Author):

The authors have responded comprehensively to my previous comments. There are interesting differences between the % of FRET negative cells, VAD-FLICA-positive, and annexin V-positive cells, which could indicate different modes of cell death at play, and these have been documented in the supplemental data which I feel is useful. I have no further comments for improvement.

We thank the Reviewers for their time, thoughtful comments and helpful suggestions that have improved our manuscript.